# ◀●▶Proppo: a Message Passing Framework for Customizable and Composable Learning Algorithms

**Paavo Parmas**[*]
Kyoto University
paavo@sys.i.kyoto-u.ac.jp

**Takuma Seno**[†]
Keio University
seno@ailab.ics.keio.ac.jp

## Abstract

While existing automatic differentiation (AD) frameworks allow flexibly composing model architectures, they do not provide the same flexibility for composing learning algorithms—everything has to be implemented in terms of back-propagation. To address this gap, we invent Automatic Propagation (AP) software, which generalizes AD, and allows custom and composable construction of complex learning algorithms. The framework allows packaging custom learning algorithms into propagators that automatically implement the necessary computations, and can be reused across different computation graphs. We implement Proppo, a prototype AP software package built on top of the Pytorch AD framework. To demonstrate the utility of Proppo, we use it to implement Monte Carlo gradient estimation techniques, such as reparameterization and likelihood ratio gradients, as well as the total propagation algorithm and Gaussian shaping gradients, which were previously used in model-based reinforcement learning, but do not have any publicly available implementation. Finally, in minimalistic experiments, we show that these methods allow increasing the gradient estimation accuracy by orders of magnitude, particularly when the machine learning system is at the edge of chaos.[3]

## 1 Introduction

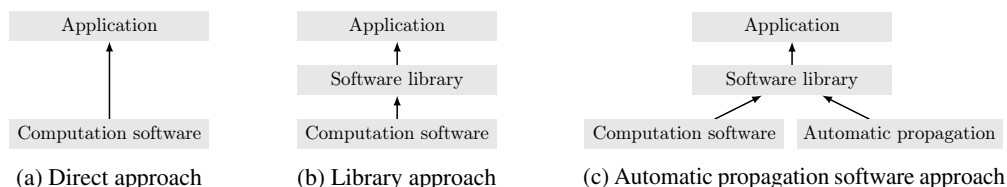

(a) Direct approach  (b) Library approach  (c) Automatic propagation software approach

Figure 1: A simplified taxonomy of machine learning (ML) software development methodologies. The *computation software* refers to existing ML tools such as PyTorch (Paszke et al., 2019) or NumPy (Harris et al., 2020); the *software library* is reusable code for a new ML method.

When a machine learning (ML) researcher comes up with a new algorithm, they are faced with a dilemma: do they directly implement an application of their algorithm using existing ML tools (Fig. 1a); or do they implement a library for their algorithm, then use the library to implement the desired application (Fig. 1b). The first approach is faster, because the researcher only has to worry about the details of the specific application; the second approach may take longer to implement, but will enable reuse of the code and speed up implementing future applications. Clearly, the second

---

[*]Affiliated with the Okinawa Institute of Science and Technology (OIST) during much of the project.

[†]Hired by OIST during his contribution to the project. Currently working at Sony AI.

[3]The code is available at `https://github.com/proppo/proppo`.

36th Conference on Neural Information Processing Systems (NeurIPS 2022).

approach is preferable in the long-term. Our work aims to overcome this dilemma by simplifying the intermediate stage of implementing reusable code for the algorithm. The approach is illustrated in Fig. 1c, where, our invention, *automatic propagation software*, ameliorates the shortcomings of existing tools that we utilize concurrently to implement reusable code for a new algorithm. This interusability was a key design choice, as it allows benefitting from the rich ML software ecosystem.

In ML a vast amount of effort has already been placed in developing computational software related to back propagation, ranging from established automatic differentiation (AD) libraries (Paszke et al., 2019; Tokui et al., 2015; Abadi et al., 2016; Bradbury et al., 2018); newer AD libraries(Oktay et al., 2020; Paszke et al., 2021); deep learning libraries (Chollet et al., 2015); extensions to AD libraries (Dangel et al., 2020; Rogozhnikov, 2021) to frameworks that use these AD libraries as tools to implement algorithms (Krieken et al., 2021; Tran et al., 2016; Gardner et al., 2018). Moreover, some libraries do not directly build on these frameworks, but can be combined together, e.g., Ray (Moritz et al., 2018) for distributed computing. Creating each one of these libraries takes significant effort, and it makes sense to build on the work of others to simplify implementing new algorithms. Due to this, new implementations tend to follow the old design paradigms of existing software.

Barham and Isard (2019) described this situation with the sentence "Machine learning systems are stuck in a rut". They argued that current ML systems are optimized for computations encountered by popular algorithms, yet, other calculations may be slow, despite in principle requiring a similar amount of operations. Such an imbalance would discourage researchers from trying new ideas that require novel forms of computation. Our own observation is that existing AD frameworks encourage implementing algorithms by designing a differentiable "surrogate loss" (Schulman et al., 2015; Foerster et al., 2018) often employed for implementations of Monte Carlo (MC) gradient estimators (Krieken et al., 2021). However, this may not be the most natural way to express the algorithms, or indeed, it may not even be possible. Minka (2019) argued that a general-purpose ML language should "simplify implementation of *all* ML algorithms", and promoted a message passing approach.

Message passing (MP) is a promising candidate for a general ML framework because of the many already existing efficient algorithms that are based on it. Some of the prominent MP algorithms are belief propagation (Pearl, 1982) and expectation propagation (Minka, 2001) in probabilistic modeling, and turbo codes in information theory (McEliece et al., 1998). In modern deep learning, MP is used in graph neural networks (Wu et al., 2020). Also, back propagation can be seen as an MP algorithm that passes the gradient messages backwards. A closely related framework to MP is the Actor Model of computation (Hewitt et al., 1973), where programs operate by actors passing messages between each other. The Actor Model has been suggested as a promising framework for parallel AI systems (Hewitt and Lieberman, 1983) and has been noted to enable modularity (Agha, 1986). We aim to incorporate these strengths into a practical general-purpose message passing framework that enables conveniently implementing a variety of MP algorithms, and propose automatic propagation (AP).

We list our desiderata for a general-purpose framework, and the approach we take to meet these desired properties in Proppo, our prototype AP software package.

**Ease of use.** Proppo is implemented entirely in Python, a language favored by ML researchers.

**Flexibility.** Proppo allows implementing arbitrary message passing algorithms.

**Computational speed.** Proppo interfaces with existing computational frameworks such as PyTorch to achieve efficient computation. Proppo adds functionality to such frameworks; the computational speed will depend on how the user implements their algorithm.

**Reusability of code.** Proppo allows packaging algorithms into propagators—the key building blocks of AP software—that encapsulate an algorithm and can be reused across different computation graphs.

**Composability.** Proppo includes special sequence propagators (Sec. 3.3) that allow combining multiple propagators together to form new propagators. These new propagators may then be further composed, allowing to deal with arbitrary complexity.

The main functionality of Proppo is to provide base classes for the propagators, and to implement automatic message passing. When a user wants to implement a new algorithm, they need to create the propagators to represent their algorithm as a message passing program. When a user wants to use an existing algorithm, they can simply download the existing propagators, and apply them in their code.

To demonstrate the utility of Proppo, we use it to implement the total propagation (TP) and Gaussian shaping (GS) gradient algorithms (Parmas et al., 2018; Parmas, 2018). Both of these are techniques for MC gradient estimation without previously existing publicly available implementations, and they were one of our key motivations for this research. In particular, the probabilistic computation graph framework (Parmas, 2018, 2020) influenced our thinking on MC gradient estimation, as it makes clear that MC gradient estimators can be implemented as MP programs passing information backwards. Our experimental results in Sec. 4.2 and App. C.3 demonstrate orders of magnitude improvement in gradient accuracy for these methods. The improvement was particularly large when the system was at the *edge of chaos*. This implies that AP software can achieve fundamentally better computational results compared to the current AD software paradigm. To demonstrate the scalability and practicality of Proppo, in our concurrent work (Anonymous, 2023) we use Proppo to replicate the model-based RL results of (Parmas et al., 2018) and we apply TP to the high dimensional visual MBRL algorithm Dreamer (Hafner et al., 2020). All our results imply that Proppo is scalable, leads to more modular code, simplifies implementation of some algorithms and can lead to greatly improved performance.

## 2 Preliminaries

### 2.1 Automatic differentiation software

A recent review of AD in ML was written by Baydin et al. (2018). To explain the functionality of AD software, suppose that a user has written a program to compute the output of a function $y = f(\boldsymbol{x})$ given the input $\boldsymbol{x}$. In such a situation, AD software can automatically obtain the gradient $\nabla_{\boldsymbol{x}} f(\boldsymbol{x})$. Such functionality is achieved by differentiating each intermediate computation and applying the chain rule. A series of computations can be represented as a directed acyclic graph with nodes and their associated variables $\boldsymbol{x}_i$, and with an edge from node $i$ to $j$ when the variable $\boldsymbol{x}_j$ is computed with a function $f_k(\boldsymbol{x}_i, \ldots)$ taking the variable at node $i$ as an input. In this case, the total derivative from $\boldsymbol{x}$ to $y$ can be obtained by summing the product of derivatives across all paths between the two nodes (Bauer, 1974),

$$\nabla_{\boldsymbol{x}} f(\boldsymbol{x}) = \sum_{\text{Path} \in \text{Paths}[\boldsymbol{x} \to y]} \prod_{\text{Edge}[l \to k] \in \text{Path}} \frac{\partial \boldsymbol{x}_k}{\partial \boldsymbol{x}_l}. \tag{1}$$

A naïve way to implement AD would be to compute the intermediate Jacobians $\frac{\partial \boldsymbol{x}_k}{\partial \boldsymbol{x}_l}$ by differentiating the operations $f_k(\boldsymbol{x}_l, \ldots)$. However, note that this will require storing full Jacobian matrices having size $K \times L$, where $K$ and $L$ are the respective dimensionalities of the variables. Moreover, if the gradient were computed by multiplying the Jacobians in the forward direction from $\boldsymbol{x}$ to $y$, this would require matrix-matrix multiplications at each step, and may incur a large computational cost. Reverse mode automatic differentiation (also known as back propagation), overcomes this issue by performing the computations backwards from the $y$ node. As $y$ is just a 1 dimensional scalar, its local Jacobian is a vector. Therefore the computation at the last step is a vector-matrix product, yielding another vector. This vector is propagated backwards, yielding a vector at each backward step. This allows computing the total gradient using only vector-matrix products, leading to a large computational saving compared to computing the gradients in the forward direction using matrix-matrix products.

Another subtle point is that it is not necessary to compute the intermediate Jacobians $\frac{\partial \boldsymbol{x}_k}{\partial \boldsymbol{x}_l}$, one merely needs a means to compute $\boldsymbol{v}^T \frac{\partial \boldsymbol{x}_k}{\partial \boldsymbol{x}_l}$, where $\boldsymbol{v}$ is the back propagated gradient vector. So in practice, AD software often only stores the variables necessary to evaluate, $\boldsymbol{v}^T \frac{\partial \boldsymbol{x}_k}{\partial \boldsymbol{x}_l}$, and never explicitly computes $\frac{\partial \boldsymbol{x}_k}{\partial \boldsymbol{x}_l}$. This leads to further computational and memory savings, as the full Jacobian matrices with size $K \times L$ need not be stored. In PyTorch, this is implemented in the `torch.autograd.backward(tensors, grad_tensors)` function, where `tensors` corresponds to $\boldsymbol{x}_k$ and `grad_tensors` corresponds to $\boldsymbol{v}$. All in all, AD software implements *forward* functions to cache the necessary data at each node, and *backward* functions to evaluate the vector-Jacobian product at each node using the cache, resulting in efficient and automatic computation of the gradient.

### 2.2 Basics of Monte Carlo gradient estimation

In ML, the computations may be stochastic, and the objective is an expectation $J = \mathbb{E}_{\boldsymbol{x} \sim p(\boldsymbol{x}; \beta)} [f(\boldsymbol{x})]$. In this case, it is not clear how to directly differentiate the stochastic operations, and we use

Monte Carlo (MC) methods to obtain an estimator, $\hat{g}$, for the gradient $\frac{dJ}{d\beta}$, s.t. $\mathbb{E}[\hat{g}] = \frac{dJ}{d\beta}$. Such methods have been reviewed in ML by Mohamed et al. (2020). The two major MC gradient estimators are the reparameterization (RP) (Rezende et al., 2014; Kingma and Welling, 2013) and likelihood ratio (LR) (Williams, 1992; Glynn, 1990) gradient estimators. The LR method uses $f(x)$ to construct an estimator, whereas RP uses $\nabla_x f(x)$. Total propagation (TP) (Parmas et al., 2018) combines LR and RP using inverse variance weighting. Moreover, TP does so at a step-wise level at each sampling node. The required step-wise computations interfere with the natural operation of back propagation, necessitating AP software to implement TP. We give more background on these methods in App. A.

## 3 Proppo explanation

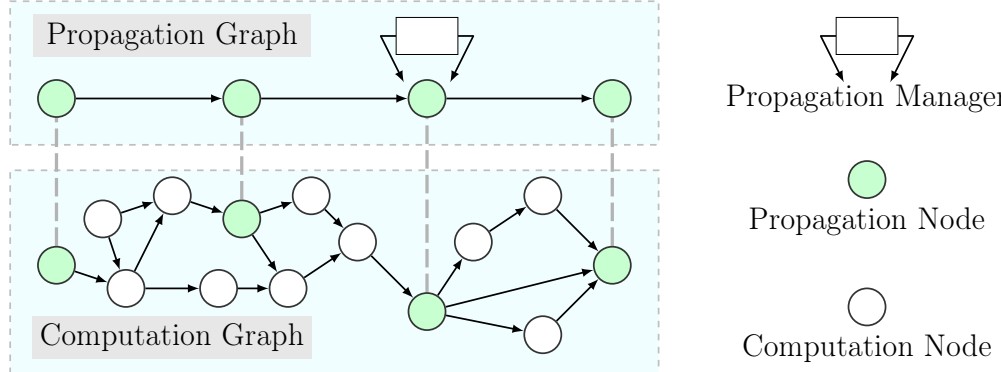

Figure 2: Automatic propagation framework (Sec. 3). A subset of the nodes ○ are designated as propagation nodes ○ forming the propagation graph. The propagation manager ▽ traverses the propagation graph and activates the propagators to ensure the correct functioning of the implemented algorithm. Here, the propagation graph is illustrated as a chain, but it may also have a graph structure. The gray dashed lines show the corresponding same nodes in the propagation and computation graphs.

A schematic illustration of automatic propagation software is provided in Fig. 2. Automatic propagation is *not* an alternative to existing frameworks, such as AD packages, rather, it adds functionality on top of them, and can be used in conjunction. We first explain the Proppo framework, in its simplest form, which is one specific implementation of AP software; we explain the different components of the framework (Sec. 3.1), then the processes by which they operate (Sec. 3.2). Finally, we explain AP from a more general perspective (App. B.1), loosening some of the specific choices in Proppo.

### 3.1 Components of Proppo

**Computation nodes.** Automatic propagation software can be combined together with any other computation package or program code. These external packages can be used regularly as one would usually write code. However, at certain locations in the code, the user can specify special computations that will determine what algorithm is implemented using the AP software. These special computations will create *propagation nodes* ○.

**Propagation nodes.** The propagation nodes determine the structure of the operations performed in the AP software. In Proppo, each propagation node includes the three components: (1) a pointer to a propagator ·○· that determines the forward and backward computations belonging to the node, (2) a cache/memory that stores necessary information for the computations occurring at the node, (3) a message box for incoming messages ⊠ from other propagation nodes.

**Propagators.** The propagators are the key components of AP software. The propagators determine the computations performed at each propagation node. In Proppo, each propagator implements a forward and backward mode of computation. The forward function is used when the node is first created; it may take an input from the program code, perform computations, store data in the memory of the node, and produce an output. The backward function is used in later processing; it performs computations using the stored data in the memory and incoming messages to the node, and sends messages ⊠ to other nodes in the propagation graph.

✉ **Messages.** Globally, the algorithms are implemented by passing messages between the propagation nodes. Each message includes the two components: (1) the contents of the message, (2) an address determining where the message should be sent. A typical address may be a pointer to the parent node of the current propagation node, so that the messages are passed backwards.

▽ **Propagation manager.** In Proppo, all of the messages are first passed to the propagation manager, then the propagation manager decides where to send the message based on the address in the message. The propagation manager also has other important roles: it constructs the propagation graph, keeps track of the nodes, activates the propagators and passes messages.

### 3.2 Typical program flow and usage of automatic propagation software

Usage of AP software can be broadly categorized into two: (1) implementing a new algorithm into an AP package, (2) using an algorithm already implemented in AP software. We start by explaining how one would typically use an already existing algorithm packaged into AP software. We begin by explaining a minimalistic code snippet in Fig. 3 of MC gradient estimation with Proppo.

```
1
2
3
4
5 state = torch.randn(batch_size,
6                     num_dimensions)
7 for t in range(horizon_length):
8     state = rnn(state)
9
10
11 loss = loss_func(state)
12 loss.backward()
```

```
1 # Configuration code
2 prop = RPProp(config) # Choose RPProp or LRProp
3 manager = PropagationManager(
4                     default_propagator=prop)
5 # Program code
6 state = torch.randn(batch_size, num_dimensions)
7 for t in range(horizon_length):
8     state = rnn(state)
9     state = manager.forward(state)
10
11 loss = loss_func(state)
12 manager.backward(loss)
```

     (a) PyTorch code         (b) Proppo code combined with PyTorch code

Figure 3: Code for estimating the gradient w.r.t. the parameters of a recurrent neural network (`rnn`).

**Minimalistic code example.** In the example in Fig. 3a, a batch of initial states (`state`) is sampled from a Gaussian distribution (lines 5–6), this batch is pushed through the recurrent neural network (`rnn`) for `horizon_length` times, then a loss is computed using the function `loss_func`, finally the gradients are backpropagted from the loss node by calling `loss.backward()`. As explained in the preliminaries (Sec. 2.1), the gradients will backpropagate until the parameter node in the computation graph. The regular PyTorch code in Fig. 3a computes the gradient using back propagation; the Proppo code in Fig. 3b modifies the computation by adding noise into the forward computation, and estimating the gradient using either RP gradients or LR gradients. Conveniently, only the definition of the propagator on line 2 has to be changed to switch between using LR or RP gradients. Moreover, settings such as whether the LR gradient should use a baseline or not can be specified in the `config` without having to modify the program code. Finally, note that instead of changing the behavior to use MC gradient estimators by using `RPProp` or `LRProp`, one could also simply use standard back propagation by setting an appropriate different propagator in the configuration. This was a minimalistic example to give a taste of how Proppo might be used. In practice both the configuration and program code may be more complicated with multiple types of propagators. We will explain how existing AP algorithms would typically be used with reference to this minimalistic example.

**Using an AP algorithm.** The code for using AP software can be broadly separated into two: the *configuration code* that sets up the AP software and selects the algorithm to be used, and the *program code* that determines how and where the algorithm is applied. It may be beneficial to keep these sections of the code distinct to improve the readability and modularity.

In the minimalistic example in Fig. 3b the *configuration code* is on lines 2–4. When configuring the code, the user instantiates the propagator and propagation manager objects from their existing class code, while giving them the appropriate configuration. A propagator may be created using

$$\text{propagator = Propagator(config).}$$

There are different propagator classes, and the user would use a different command for the different types of propagators. Some propagators may allow for a configuration input to make small tweaks to

the behavior. The propagation manager may be created using

$$\texttt{manager = PropagationManager(config)}.$$

Here the configuration input may define default propagators to be used for specified computations, default message passing styles, etc. The configuration in the RNN example was simple, but in general the configuration may be more complicated and include setting up many different types of propagators and equipping them to the manager by assigning them names by which the manager identifies them.

After selecting the desired algorithm by properly configuring the AP software, the user must modify the *program code* to make it compatible with the AP software and properly implement the algorithm. In the minimalistic example in Fig. 3b, the program code is on lines 6–12, and the modifications to use Proppo are on lines 9 and 12. We emphasize that the same modifications may be compatible with multiple algorithms, so that only the configuration code has to be modified to switch between the algorithms. The main purpose of the modifications to the program code are to activate the propagators and construct the propagation graph. A propagator is activated using the command

$$\texttt{output = manager.forward(input, arguments)}.$$

This command will add a node into the propagation graph of the manager. In the current case, the `manager` has been equipped with the default propagator `prop`. When `manager` calls the forward function, it will create a new node, and assign the `prop` propagator to it. Then, the forward method of `prop` is called with the `input`. The forward method applies some modification to the input, stores data in the memory of the node, and produces an output. Note that the data stored in the memory of the node will be used later during the backward pass of the algorithm.

The `arguments` input to the `forward` function serves multiple purposes. One of the main purposes is to modify the behavior of the propagator in some way. For example, in the minimalistic code in Fig. 3b the propagator injects noise into `state` to produce the output, so one part of the `arguments` may be the type of noise and the parameters of the noise distribution. Note that often these settings can be defined in the configuration without having to modify the `arguments`, e.g. the type of distribution may just be fixed in the configuration code. Another role of the `arguments` is to modify the targets where the messages should be sent if the default setting is unsuitable. Furthermore, if multiple types of propagators are necessary to implement the algorithm, the `arguments` may also be used to select between different propagators equipped to the manager.

Finally, the backward pass of the algorithm is invoked by calling

$$\texttt{manager.backward()}.$$

When the `manager.backward` function is called, the manager will typically call the `backward` function of the propagator of the last node, pass the message to its target node, call the `backward` function of the previous node, and iterate this process from the end of the graph until the beginning. If the propagators are correctly implemented, and the propagation graph is correctly constructed, this framework allows automatically using an algorithm available in the AP software.

A keen reader may have noticed that in the minimalistic example in Fig. 3b an additional `loss` input was given to the `backward` call. This is a convenience implemented in Proppo, where the manager will automatically add a propagator corresponding to the loss as the final node before starting to iterate backwards on the propagation graph. Alternatively, the same operations could have been implemented by the commands `manager.append_loss(loss)` followed by `manager.backward()`. Internally, `append_loss(·)` simply calls the forward method of a special propagator designed for adding a node corresponding to a loss. In Proppo, yet another way how the lines 11–12 for adding the loss could have been implemented is the following: `manager.append_loss(state, lossfunc=loss_func)` followed by `manager.backward()`. In this case, the loss function `loss_func` is passed to the propagator directly via the `lossfunc` keyword, and the propagator will compute the loss by inputting `state` into `loss_func`. The last method of implementation is beneficial for compatibility between different types of loss propagators, for example, it is useful for easily switching between the loss operations necessary for Gaussian shaping gradients and a regular loss computation. We see that there are many different ways to implement the same algorithm using propagators.

**Creating an AP algorithm.**    In the previous section, we saw that the functionality of an AP algorithm is determined by the implementation of the propagators. Consequently, creating a new AP algorithm requires implementing the appropriate propagators corresponding to the desired algorithm.

In Proppo, several base classes for new propagators are already provided, and the user must only implement the `forward` and `backward` methods of the propagators.

In principle, the `forward` and `backward` methods can contain arbitrary code, and there is no restriction on what they can do. The programmer must figure out how to implement their algorithm through a process of storing data in the `forward` pass, and further computations and message passing in the `backward` pass. The programmer must also decide how many different types of propagators are necessary for their algorithm, and also how they are intended to be used in the *program code*. Like with any software, good design of the code and clear documentation are crucial for usability.

Throughout our discussion of AP software, we have promised composability and customizability of the algorithms. We note that merely using AP software to implement an algorithm does not guarantee that the algorithm will be composable. It is up to the creator of the algorithm to design the propagators so as to facilitate such desirable properties. However, we find that the paradigm of propagators is a good framework by which to construct an algorithm in a composable way. In particular, in Sec. 3.3 we highlight common propagator types that can lead to composable and flexible algorithms.

### 3.3 Example elementary propagators

In AP software, composability is achieved by deriving new propagators from already existing propagators. The simplest form of this may be to connect propagators into a chain, and activate their methods in sequence. In Proppo, we provide two propagators with such functionality, `ComboProp` and `ChainProp`. While both of these methods activate the forward and backward methods of the constituent propagators sequentially, they differ in how the messages and nodes are handled.

**ComboProp.** This propagator is a simple way to connect propagators into a chain. The output of the forward method of one propagator is fed into the input of the next propagator, and each propagator operates on the same node. Likewise, the backward methods use the data in the same node, and the messages are simply combined together and passed to the backward method of the previous propagator. `ComboProp` is suitable for simple combinations of propagators; however, it can run into problems when trying to implement more complex behaviors. Namely, the different propagators may overwrite data stored by other propagators leading to unexpected consequences.

**ChainProp.** This propagator provides a more systematic way to combine propagators, so as to overcome the sometimes unpredictable behavior of `ComboProp`. To rigorously define the behavior, `ChainProp` internally creates its own propagation graph, connecting nodes with propagators into a chain, and assigning a propagation manager to handle the message passing. The constituent propagators may themselves also be of the `ChainProp` type. This way, propagation graphs can be embedded into other propagation graphs, creating a hierarchy of connected propagators all handled by their own managers. This system allows safely composing propagators to define complex behaviors.

**BackPropagator.** This example propagator is often composed with a *base propagator* using `ComboProp`. In Pytorch, the `torch.autograd.backward(tensors, grad_tensors)` method allows inputting lists of tensors and gradients, so that the backpropagation can be commenced at multiple tensors at the same time using a single call. In the composition, the *base propagator* produces a tensor and the gradients, and `BackPropagator` commences the back propagation on the AD computation graph. Importantly, usually the config of the propagator allows determining whether `BackPropagator` should be included in the combo sequence or not. This is useful, if one wants to omit immediately backpropagating gradients in favor of grouping multiple tensors together and initiating each backprop simultaneously with a single call.

### 3.4 Example Monte Carlo gradient estimator propagators

One of our main motivations for developing AP software was to enable easy implementation of advanced MC gradient estimation methods, such as total propagation or Gaussian shaping gradients. We provide detailed pseudo code for these propagator implementations in App. B.2. Here, we briefly outline some of the more prominent propagator implementations.

**Reparameterization.** In PyTorch, RP gradients are already implemented, and it is possible to obtain reparameterized samples from a distribution, so that the gradients automatically flow through. Alternatively, the output can be detached, and the gradients passed backwards using `BackPropagator`.

**Likelihood ratio.** Typically, LR gradients are implemented in AD software by constructing a surrogate loss (Schulman et al., 2015) of the form $\log p(x; \theta) f(x)$, and then differentiating this w.r.t. $\theta$ to obtain the LR gradient. In Proppo, we do not need to construct such a surrogate loss. We instead store $\log p(x; \theta)$ in the node as a tensor, then during the backward pass, we set $f(x)$ as the input gradient, and back propagate it into $\log p(x; \theta)$, using the `BackPropagator` in Sec. 3.3.

**Total propagation.** TP requires storing both data for RP and LR in its node. In addition, it requires setting a target tensor in the AD computation graph where the variances of LR and RP will be computed for the inverse variance weighting. We implement this by storing a pointer to the target tensor, and use `torch.autograd.grad` to compute the LR and RP gradients. After computing the gradient variances, the mixture weight can be computed, and `BackPropagator` from Sec. 3.3 is used for one last back propagation with the combined gradient.

### 3.5 Implementation and computation speed considerations

Primarily, Proppo does not perform any major computations on its own; it merely manages the operations performed by other computational tools (Sec. 3). This overhead is negligible for moderate size computations as confirmed by benchmarking experiments (App. B.3) and real applications (Anonymous, 2023). Experiments also confirmed that Proppo can scale to millions of propagation nodes (App. B.4). The scalability could be further increased by running multiple propagation managers on the propagation graph in parallel, and we plan to implement this in future work.

## 4 Chaotic net minimalistic experiments for comparing gradient estimators

In this section we show minimalistic easily reproducible experiments using the Proppo framework. The main aim is to demonstrate that there exist situations where TP and GS gradients can improve the gradient accuracy by multiple orders of magnitude compared to conventional gradient estimation methods. The main result for TP is in Fig. 4, while the experiments for GS are in App. C.3.

### 4.1 Setup: gradient variance of a recurrent neural network with chaotic behavior

In our experiments, we estimate the gradient, $g$, of an expected loss $\mathbb{E}_{p(\boldsymbol{x}; \beta)}[f(\boldsymbol{x})]$, w.r.t. a parameter $\beta$, using various different MC gradient estimation methods. We repeat this estimation many times, and compute the empirical variance of the gradient estimators, $\hat{\mathbb{V}}[g]$, from the set of estimates $\{g^{(k)}\}_{k=1}^{K}$. In terms of the particular situation we consider, we were motivated by PIPPS (Parmas et al., 2018), where they showed a situation where chaos-like properties of the dynamics caused the RP gradient variance to explode, while LR gradients were robust. Parmas et al. (2018) considered a cart-pole swing-up and balancing task; while this task appears simple, there is still redundant complexity as the control policy had a high-dimensional parameter. Moreover, while the fractal input-output patterns that they visualized are strongly suggestive that the system was chaotic, it falls short of a rigorous proof. Therefore, we devised a similar simplified system based on a recurrent neural network (RNN).

We consider a particular 2-dimensional sigmoid recurrent neural network that was mathematically proven to exhibit chaotic behavior by Wang (1991). Formally, the system dynamics are given by

$$\boldsymbol{x}_{t+1} = \text{Sigmoid}(\beta W \boldsymbol{x}_t) + \varepsilon, \tag{2}$$

where $\text{Sigmoid}(y) := \frac{1}{1+\exp(-y)}$, $W = \begin{bmatrix} -5 & 5 \\ -25 & 25 \end{bmatrix}$, $\beta$ is the inverse temperature, a variable parameter of the network, $\varepsilon \sim \mathcal{N}(0, I\sigma^2)$ is Gaussian noise added to the state at each time step,[4] $\sigma = 0.001$ is the standard deviation of the added activation noise, and the initial state is $\boldsymbol{x}_0 = [0.35; 0.55]$. We use a batch size, $B = 1000$, and simulate this system for $H$ time steps. We compute a quadratic loss at the final time step $L(\boldsymbol{x}_H) = \frac{1}{2}(\boldsymbol{x}_H - \boldsymbol{1})^{\mathsf{T}}(\boldsymbol{x}_H - \boldsymbol{1})$ and estimate the influence through all time steps to obtain the gradient w.r.t. $\beta$. Wang (1991) explained that $\beta$ controls the dynamics of the system. When $\beta$ is small, the system is well-behaved, but as $\beta$ is increased, the system becomes chaotic through a period-doublings mechanism. We discuss this system in greater depth in App. C.1.

---

[4]Note that Wang (1991) did not add any noise to the system, but the system still shows similar behavior even when such noise is added. We added this noise to be able to use LR gradients.

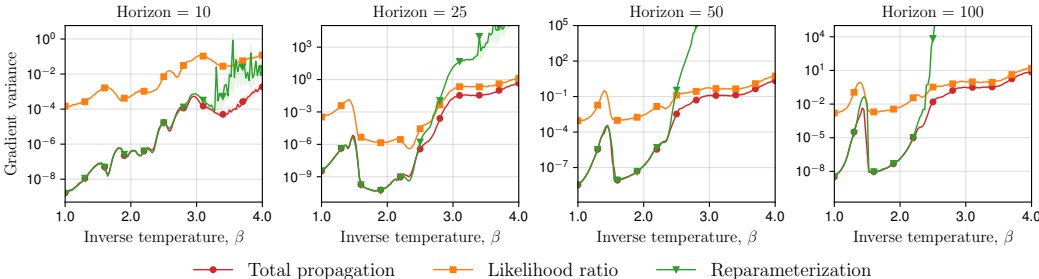

(a) Variance of RP, LR and TP gradient estimators against the inverse temperature $\beta$ for various horizons.

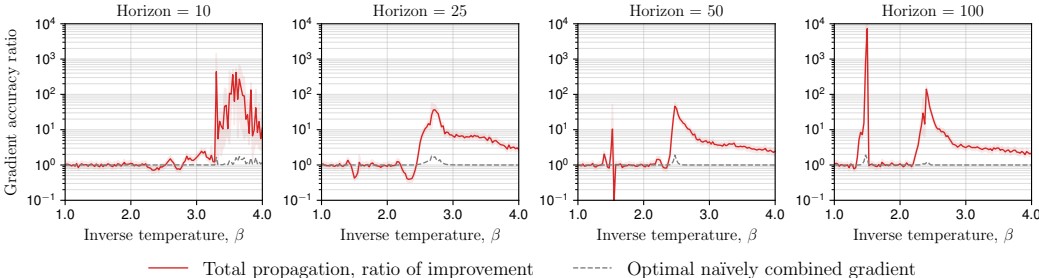

(b) Improvement of TP over the best between LR and RP for each $\beta$.

Figure 4: Comparison of the RP, LR and TP gradient estimators on the chaotic RNN task. See Sec. 4.1 and Fig. 9 for the description of the task, and Sec. 4.2 for the discussion of the experiment. The main result is that TP provides a large improvement at the edge of chaos regime around $\beta \sim 2.5$.

## 4.2 Single path experiment: showing the advantage of total propagation

**Experimental setup.** We consider an experiment on the simple RNN system (Sec. 4.1). We compute the gradient variances of RP, LR and TP gradients, and plot these against $\beta$. We considered the horizons $H \in \{10, 25, 50, 100\}$. The batch size was $B = 1000$ for 1 gradient estimate, $g$ (averaged over the batch), and we performed this estimated $K = 800$ times. We compute the empirical gradient variance of the $K$ estimates, $\{g^{(k)}\}_{k=1}^{K}$. We use the percentile bootstrap method to estimate 95% confidence intervals. The results are in Fig. 4.

**Results.** The gradient variances are plotted in Fig. 4a. We see that RP is much more accurate than LR at low $\beta$, but the gradient variance of RP explodes as the system transitions to chaos around $\beta = 2.5$. This effect is stronger as the horizon is increased. Fig. 4b shows the ratio of improvement of TP over the best between RP and LR at each $\beta$. We see that around the edge of chaos at $\beta = 2.5$ TP improves by up to around 100 times compared to the best choice between RP and LR.

**Discussion.** Previously (Parmas et al., 2018) already showed an example where the RP gradient variance explodes. They also showed that LR was robust to this issue, and demonstrated that TP improves the performance. However, in their results, while the improvement of both TP and LR was huge compared to RP in the chaotic regime, the improvement of TP over LR was comparatively small, up to around a 3 times improvement. Here, on the other hand, we showed that TP can simultaneously improve over both RP and LR by multiple orders of magnitude. We also elucidated that the improvement may be particularly large at the edge of chaos. The simplicity of our RNN example allowed examining the advantage of TP in greater depth with better reproducibility.

In Fig. 4b, we have additionally plotted the performance for an optimal "naïve" gradient combination based on applying inverse variance weighting to LR and RP computed separately,[5] without taking advantage of the graph structure of the computations. Such a naïve method was first mentioned by Parmas et al. (2018) who immediately skipped it for TP, because of its conceptual issues; however,

---

[5]Strictly speaking, we did not actually combine the gradient estimates, but only computed the variance of an idealized version of this estimator by using the precise variances $\mathbb{V}[g_{\text{RP}}]$ and $\mathbb{V}[g_{\text{LR}}]$. In practice, one also has to estimate these variances and the mixing ratio from each batch of samples.

later Metz et al. (2019) "followed the insight" from Parmas et al. (2018) and used the naïve method in the context of metalearning. In Fig. 4b, we see that the naïve method performs poorly, leading to hardly any improvement in gradient accuracy compared to the best between RP and LR.

Indeed, when combining statistical estimators, the theoretical maximum accuracy is the sum of the accuracies of the individual estimators. If two estimators are equally good, this would maximally lead to a doubling of the accuracy. However, in Fig. 4b we see improvements far exceeding this limit. Such a result was possible because TP takes advantage of the graph structure of the computations. It not only improves the accuracy of the final result, it also improves the accuracy of the intermediate gradient computations that are propagated backwards. The improvements in the accuracy of the intermediate calculations have a compounding effect, leading to orders of magnitude improvement of the final result. This advantage of TP highlights that automatic propagation software such as Proppo can produce results unattainable by standard back propagation procedures.

## 5 Related work

From the point of view of MC gradient estimation, the concurrent work, Storchastic (Krieken et al., 2021) is the most closely related to ours. Storchastic is a library for MC gradient estimation implemented in PyTorch based on the surrogate loss formalism (Schulman et al., 2015; Foerster et al., 2018). Their user interface of using a `.backward()` function is similar to ours, as both were inspired by the PyTorch interface that is the same. However, Storchastic is not an AP software, and for instance, it does not enable implementing Total Propagation.

From a conceptual point of view, the Actor Model (Hewitt et al., 1973) is related. In the Actor Model, everything in the program are actors and they communicate with each other via message passing. However, practically, Proppo is not implemented as an actor system. Unlike in an actor system, the propagators acting at the nodes can interact with outside objects directly without explicitly implemented messaging mechanisms. From a theoretical point of view, one could consider these interactions as "passing messages" but practically, these interactions are not implemented based on the actor formalism. Instead, AP software aims to be pragmatic and flexible to enable interusability with existing ML tools. A further difference is that AP automates the default messaging targets. Prominent Actor Model implementations, such as Akka (Bonér et al., 2010) are typically used for server and web apps, and they are not primarily meant for implementing machine learning algorithms. Nevertheless, studying existing actor systems may inspire improvements to AP implementations.

## 6 Discussion & Conclusions

Recently, many impressive results were achieved by leveraging large scale data and computation together with basic algorithms (Brown et al., 2020; Ramesh et al., 2022; Chowdhery et al., 2022). We ask whether simple algorithms are used, simply because researchers and engineers have difficulty with properly implementing more complicated ones? There is no guarantee that merely scaling up our existing algorithms will be sufficient for satisfactory performance in the long term of ML research; we must also explore algorithmic improvements. We hope that automatic propagation will enable the proliferation of complex algorithms by providing a standard for sharing and building ML tools.

**Author Contributions**

Paavo Parmas invented automatic propagation software, designed and implemented most of the code including the key parts, ran the experiments, and wrote the paper.
Takuma Seno worked part-time to help write the code and give advice on the software engineering aspects in the early stages of the project. He also gave comments on the paper.

**Acknowledgements**

The project was started with funding provided by the Proof of Concept Program at the Okinawa Institute of Science and Technology (OIST). Most of the key contributions were made during this stage. PP finished the project while hired under the Cyborg-AI project supported by NEDO, Japan. PP would like to thank the Neural Computation Unit at OIST for administrative support while the project was being conducted at OIST.

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
