# Appendices: ◄O►Proppo: a Message Passing Framework for Customizable and Composable Learning Algorithms

## A Monte Carlo gradient estimators

### A.1 Basic Monte Carlo gradient estimation

In this section we provide additional background explanations on the two main gradient estimators: reparameterization gradients (RP) and likelihood ratio gradients (LR). Both are MC gradient estimation methods, i.e., they provide estimators, $\hat{g}$, s.t. $\mathbb{E}\left[\hat{g}\right] = \frac{d}{d\beta}\mathbb{E}_{\boldsymbol{x}\sim p(\boldsymbol{x};\beta)}\left[f(\boldsymbol{x})\right]$.

**RP.** In RP, one samples from a simple fixed distribution $\varepsilon \sim p(\boldsymbol{\varepsilon})$, and one defines a transformation, $\mathcal{T}$, dependent on $\beta$ s.t. $\mathcal{T}(\boldsymbol{\varepsilon};\beta)$ has the same distribution as a sample from the original distribution $\boldsymbol{x} \sim p(\boldsymbol{x};\beta)$. Then, the derivative can be pushed inside the expectation, and the gradient can be estimated as

$$\frac{d}{d\beta}\mathbb{E}_{\boldsymbol{x}\sim p(\boldsymbol{x};\beta)}\left[f(\boldsymbol{x})\right] = \mathbb{E}_{\boldsymbol{\varepsilon}\sim p(\boldsymbol{\varepsilon})}\left[\frac{d}{d\beta}f\big(\mathcal{T}(\boldsymbol{\varepsilon};\beta)\big)\right] = \mathbb{E}_{\boldsymbol{\varepsilon}\sim p(\boldsymbol{\varepsilon})}\left[\frac{df}{d\boldsymbol{x}}\frac{d\mathcal{T}(\boldsymbol{\varepsilon};\beta)}{d\beta}\right]. \tag{3}$$

For example, for a 1-dimensional Gaussian distribution $\mathcal{N}(x;\mu,\sigma)$, one possible reparameterization is $\mathcal{T}(\varepsilon;\mu,\sigma) = \mu + \varepsilon\sigma$, where $\varepsilon \sim \mathcal{N}(\varepsilon;0,1)$, and then $\frac{d\mathcal{T}}{d\mu} = 1$ and $\frac{d\mathcal{T}}{d\sigma} = \varepsilon$.

**LR.** While RP used the gradient of $f$ to construct an estimator, LR, instead, uses its value. The LR gradient estimator is characterized by the equation

$$\frac{d}{d\beta}\mathbb{E}_{\boldsymbol{x}\sim p(\boldsymbol{x};\beta)}\left[f(\boldsymbol{x})\right] = \mathbb{E}_{\boldsymbol{x}\sim p(\boldsymbol{x};\beta)}\left[\frac{d\log p(\boldsymbol{x};\beta)}{d\beta}(f(\boldsymbol{x}) - b)\right], \tag{4}$$

where $b$ is a baseline for variance reduction, often computed as the batch mean of the samples $b = \frac{1}{B}\sum_{i=1}^{B}f(\boldsymbol{x}^{(i)})$.

Both LR and RP estimators are interchangeable (as long as the gradients and value of $f$ are available), and which one is better depends on the specific situation. While RP tends to handle high dimensional spaces better (Rezende et al., 2014), it often fails on long computation graphs (Parmas et al., 2018). On the other hand, LR has robust behavior, and does not require access to $\nabla f$, but does not scale well with the dimensionality. The two estimators were recently unified by Parmas and Sugiyama (2021) based on an intuititve probability flow theory related to the work of Jankowiak and Obermeyer (2018).

A key metric to determine which gradient estimator is effective is the *variance* of the gradient estimator, $\mathbb{V}\left[\hat{g}\right]$. The variance can be reduced by a factor $1/K$ by computing $K$ samples of the same estimator and averaging; therefore, we could consider a reduction in the gradient variance by a factor $K$ to be roughly equivalent to increasing the computation speed by a $K$ factor.[6] For this reason, much research on MC gradient estimators has focused on reducing the variance, primarily by using control variates and baselines (Greensmith et al., 2004; Weaver and Tao, 2001); or conditioning and importance sampling (Owen, 2013). Another line of research takes advantage of the graph structure of the computations to obtain more accurate gradient estimates (Parmas et al., 2018; Parmas, 2018, 2020). While the former methods are readily implemented using the surrogate loss formalism (Schulman et al., 2015), the latter methods are not easily implemented, motivating the creation of Proppo.

In this section, we discussed how the basic estimators are implemented through a single sampling operation. In the next section, we discuss graphs with multiple stochastic operations, and also introduce the total propagation and Gaussian shaping gradient methods, which take advantage of this graph structure.

### A.2 Monte Carlo gradient estimation on Probabilistic Computation Graphs

In the deterministic case, the total derivative intuitively decomposes into a sum across the paths as shown in Eq. (1); Parmas (2018) explained that a similar framework can be employed for stochastic graphs, using their probabilistic computation graph (PCG) formalism. In a PCG, we note that while the sampling operations themselves are *stochastic*, the relationship between the marginal distributions

---

[6]Note that in practice parallel computation may allow increasing the batch size without proportionally increasing the computation time, if the computational resources are not already maxed out. Moreover, reducing the gradient variance by a factor $K$ may not guarantee proportionally faster optimization because increasing the gradient accuracy has diminishing returns once the gradient is sufficiently accurate.

at nodes is *deterministic*. Therefore, by denoting $\zeta_{\boldsymbol{x}_i}$ as the abstract parameters of the marginal distribution at the node $i$, and replacing the usual derivative with functional derivatives between the probability distributions, we can obtain a similar decomposition of the total gradient as in Eq. (1)

$$\frac{\mathrm{d}\zeta_y}{\mathrm{d}\beta} = \sum_{\text{Path}\in\text{Paths}[\boldsymbol{\beta}\to y]} \prod_{\text{Edge}[l\to k]\in\text{Path}} \frac{\partial\zeta_{\boldsymbol{x}_k}}{\partial\zeta_{\boldsymbol{x}_l}} \tag{5}$$

Parmas (2018) proposed a further decomposition by assigning a set of intermediate nodes $\mathcal{N}$, and considering the paths passing through the nodes $n \in \mathcal{N}$ giving the equation

$$\frac{\mathrm{d}\zeta_y}{\mathrm{d}\beta} = \sum_{n\in\mathcal{N}} \left( \sum_{\text{Path}\in\text{Paths}[n\to y]} \prod_{\text{Edge}[l\to k]\in\text{Path}} \frac{\partial\zeta_{\boldsymbol{x}_k}}{\partial\zeta_{\boldsymbol{x}_l}} \right) \left( \sum_{\text{Path}\in\text{Paths}[\beta\to n]\setminus\mathcal{N}} \prod_{\text{Edge}[l\to k]\in\text{Path}} \frac{\partial\zeta_{\boldsymbol{x}_k}}{\partial\zeta_{\boldsymbol{x}_l}} \right), \tag{6}$$

where $\text{Paths}[\beta \to n] \setminus \mathcal{N}$ denotes the paths going from $\beta$ to $n$, but not passing through nodes in $\mathcal{N}$. Parmas (2018) further showed that Eqs. (5) and (6) can be combined to yield

$$\frac{\mathrm{d}\zeta_y}{\mathrm{d}\beta} = \sum_{n\in\mathcal{N}} \frac{\mathrm{d}\zeta_y}{\mathrm{d}\zeta_n} \left( \sum_{\text{Path}\in\text{Paths}[\beta\to n]\setminus\mathcal{N}} \prod_{\text{Edge}[l\to k]\in\text{Path}} \frac{\partial\zeta_{\boldsymbol{x}_k}}{\partial\zeta_{\boldsymbol{x}_l}} \right), \tag{7}$$

and they explained that this equation generalizes the deterministic and stochastic policy gradient theorems (Sutton et al., 2000; Silver et al., 2014). In particular, the $\frac{\mathrm{d}\zeta_y}{\mathrm{d}\zeta_n}$ terms are estimated by total derivative estimators, such as LR or value gradient methods (Fairbank, 2008), whereas the effect of the local partial derivatives, $\frac{\partial\zeta_{\boldsymbol{x}_k}}{\partial\zeta_{\boldsymbol{x}_l}}$, is estimated by pathwise estimators, such as direct differentiation or RP. They also proposed more advanced estimators: total propagation that combines RP and LR (Parmas et al., 2018) and Gaussian shaping gradients that use a different decomposition of the paths to obtain a gradient estimator (Parmas, 2018).

**Total propagation (TP).** Inverse variance weighting is the optimal method to take a weighted average of two uncorrelated statistical estimators, as is well known in statistics. The TP method (Parmas et al., 2018) uses this weighting scheme to obtain a weighted average of LR and RP, based on the observation that both estimators are interchangeable. In particular, it performs the computation $\hat{\boldsymbol{g}}_{\text{TP}} = k\hat{\boldsymbol{g}}_{\text{LR}} + (1-k)\hat{\boldsymbol{g}}_{\text{RP}}$, where $k = \mathbb{V}[\hat{\boldsymbol{g}}_{\text{RP}}]/(\mathbb{V}[\hat{\boldsymbol{g}}_{\text{LR}}] + \mathbb{V}[\hat{\boldsymbol{g}}_{\text{RP}}])$, i.e. it picks $k \propto \frac{1}{\mathbb{V}[\hat{\boldsymbol{g}}_{\text{LR}}]}$ and $(1-k) \propto \frac{1}{\mathbb{V}[\hat{\boldsymbol{g}}_{\text{RP}}]}$. The gradient variances for computing the weights are obtained from the empirical variances of the gradient samples. Moreover, TP is not a simple combination of the two estimators computed separately on the whole graph, instead, it combines the two estimates at each sampling node, and propagates the combined gradient backwards, potentially leading to much increased accuracy (the advantage is experimentally clear in Sec. 4.2). However, this kind of gradient variance estimation during the backward computation poses problems for existing AD software, as it cannot be implemented by differentiating a surrogate loss.

**Gaussian shaping gradients (GS).** The GS method (Parmas, 2018) is interesting because it allows obtaining an LR type gradient estimator while using $\nabla f$ instead of $f$ as is usual in LR gradients. This allows better scalability with the dimensionality compared to a regular LR method, as we show experimentally in Sec. C.3. The basic idea is to assume a Gaussian density with parameters $\boldsymbol{\mu}$ and $\Sigma$ at a distal node $n$, then construct an LR-type gradient estimator for these parameters, estimating the gradients $\frac{\mathrm{d}\boldsymbol{\mu}}{\mathrm{d}\boldsymbol{x}}$ and $\frac{\mathrm{d}\Sigma}{\mathrm{d}\boldsymbol{x}}$. Assuming a cost function $c(\boldsymbol{x}_n)$ that depends on $\boldsymbol{x}_n$, we can resample points on the approximated Gaussian distribution, estimate the gradients $\frac{\mathrm{d}\mathbb{E}[c]}{\mathrm{d}\boldsymbol{\mu}}$ and $\frac{\mathrm{d}\mathbb{E}[c]}{\mathrm{d}\Sigma}$ and apply the chain rule to obtain the total gradient from $\boldsymbol{x}$ to $c$. The final algorithm resembles LR, except that the usual $f$ multiplier is replaced with a different scalar given by the dot product of some statistics of the distribution at $\boldsymbol{x}_n$ with $\frac{\mathrm{d}\mathbb{E}[c]}{\mathrm{d}\boldsymbol{\mu}}$ and $\frac{\mathrm{d}\mathbb{E}[c]}{\mathrm{d}\Sigma}$. Similarly to TP, this method is also cumbersome to implement by differentiating a surrogate loss, but can be intuitively expressed as a message passing program that can be implemented in Proppo.

# B    Automatic Propagation software: additional details and experiments

## B.1    A generalized view of automatic propagation software

In the main contents, we introduced AP software from the viewpoint of Proppo, our prototype implementation of an AP library. We did this to provide a concrete example to aid in clarity. More generally, AP software may have many more features than introduced, or on the contrary, it may also be more minimalistic than our example with Proppo. In this section we aim to clarify what type of additional features we foresee to be useful for AP software, and also to clarify what are the minimum requirements for something to qualify as an AP library.

First, we note that it may be useful for the `forward` computations to also be able to send messages. Moreover, it can be beneficial for the forward methods to be able to pass information to the propagation manager, so as to automate the choice of propagators for subsequent nodes. In fact, Proppo already implements such automation in the choice of the configuration for the loss propagators depending on what type of MC gradient estimator was previously used. This observation points toward generalizations concerning the methods of the propagators as well as the managers.

Regarding the propagators, in our discussion we limited the computations to `forward` and `backward` computations. However, there is no need for such a restriction, and in general we require propagators to implement *types* of computations, where `forward` and `backward` are just two possible types. The different types of computations can in general access and modify the memory in the node, and may send messages to other nodes. We note that the distinction between types of computations is superficial because the multiple types of computations may also be embedded into a single type of computation, where an input can be passed to switch between the different embedded computations.

Regarding the propagation manager, this was merely our choice of implementation to keep track of the propagation graph, automatically decide on the order of activating the propagators, and to deal with passing the messages. Any other method to implement such functionality would also be allowed.

In summary the crucial aspects to automatic propagation software are:

1. Nodes that can store information.

2. Propagators that can be associated to the nodes. The propagators have user programmable methods that can directly modify the information in the nodes that they are linked to, and can indirectly influence other nodes by sending them messages containing general information.

3. The methods of the propagators may directly interact with processes and information external to the nodes.

4. When a message is sent from a propagator operating at a node, if a target for the message was not explicitly specified by the user, a default target is determined.

5. It is possible to trigger the system so that the propagators at multiple nodes are automatically activated in sequence.

Regarding points 4 and 5, we envision that typically the nodes will be arranged in an acyclic directed graph structure, and the system can be triggered to traverse the graph backwards, activating the `backward` method of the propagator at each node, and sending the messages to the parent nodes. Even if the contents of a message are not intended to be used by a direct parent node, but they are meant for a node earlier in the graph, as long as this message is continually routed backwards it will eventually reach the target destination. Thus we envision that such a default setting will allow implementing a wide range of algorithms. In fact, there already exist many prominent machine learning algorithms that operate purely based on sending messages backwards locally, e.g. back propagation or belief propagation (Pearl, 1982).

## B.2    Examples of propagators and their details: customizability and composability

In this section we aim to illustrate the composability and customizability that can be achieved in AP software. These properties are achieved by using the sequence propagators described in Sec. 3.3. These propagators allow combining multiple propagators together into a sequence to compose new propagators. The forward and backward methods of the propagators will be activated in sequence.

One of our main design patterns to effectively use these sequence propagators is to create a base propagator for each non-trivial functionality that we want to implement. Then we plug these base propagators together into a chain to achieve all of the desired properties. Some of these base propagators may be reused across many different composite propagators, allowing to create compact code. For example the `BackPropagator` in Sec. 3.3 can be combined with many different MC gradient estimation propagators to initialize the back propagation of the gradients.

Another feature of the sequence propagators is that they allow optional propagators in the sequence. These optional propagators can be turned on or off using keyword arguments when instantiating the propagator from the class. Essentially, this allows creating a factory for a composite propagator with a rich configuration space. In the following, we give pseudocode and details of several MC gradient estimation propagators, their constituent propagators, and explain how they interact.

## BackPropagator

This propagator commences backpropagating the gradients at a computation node.

**Forward:**
```
pass
```

The forward method does not do anything, as this propagator is designed to be combined together with other propagators that will perform the necessary forward computations.

**Backward:**
```
get from message:  tensors, grad_tensors
call:  torch.autograd.backward(tensors, grad_tensors)
```

The backward method retrieves the tensors and gradients, and backpropagates them using AD software as explained in Sec. 2. One of the propagators that can be combined together with `BackPropagator` is `RPBase` for reparameterization gradients explained next.

## RPBase

This is the base propagator for reparameterization gradients. It injects reparameterized noise into an input variable, and allows backpropagating through this stochasticity.

**Forward:**
```
input from program code:  x
inject reparameterized noise into x
store into node:  output, detached output
```

**Backward:**
```
get grads from detached output
message:  (output, grads)
```

Note that the backward method does not backpropagate the gradients on the computation graph. It merely forms the pair of the output node and its corresponding gradient. To have this gradient be backpropagated as well, the following compound `RPProp` propagator can be used.

## RPProp

This propagator applies the reparameterization transformations and also automatically commences the backpropagation on the computation graph.

**Sequence:**  `[Optional(BackPropagator), RPBase]`

Here we used the `Optional()` notation to mean that the `BackPropagator` can be either included or omitted using a keyword, i.e., `RPProp(backprop=False)` would omit the `BackPropagator`. If it is included, then having simply plugged these two propagators together causes the gradients to be backpropagated on the computation graph as well—`RPBase` will form the (`tensors`, `grad_tensors`) pair, and `BackPropagator` will commence the back propagation.

## LossBase

In many machine learning tasks, there are loss nodes that we want to propagate gradients from. Moreover, for likelihood ratio gradient estimators, we may also want to send the value of the loss itself backwards as well. This propagator implements the required base functionality.

**Configuration:**    whether gradients are needed or not

**Forward:**
```
compute loss
store loss in the node
```

**Backward:**
```
get from message:  incoming sum of losses
optional:  create tensors and grad_tensors for the loss
optional:  sum new loss with incoming losses
message:  [(tensors, grad_tensors), sum of losses]
```

---

## LossProp

To add automatic gradient backpropagation, or baseline computations to the loss node, we create a compound propagator with the following sequence.

**Sequence:**    [Optional(BackPropagator), Optional(BaselineProp), LossBase]

---

## BaselineProp

In LossProp, one part of the sequence was the baseline propagator. The baseline propagator subtracts a baseline from the sum of the loss to reduce the likelihood ratio gradient variance.

**Configuration:**    the type of baseline function to use, e.g., subtract the mean

**Forward:**
```
pass
```

Like the BackPropagator, the baseline propagator is also designed to be used together with other propagators, and it does not need its own forward method.

**Backward:**
```
compute baseline
subtract baseline from losses
message:  loss with baseline subtracted
```

---

## LRBase

This propagator implements the base functionality for using likelihood ratio gradient estimators.

**Forward:**
```
input from program code:  x
inject noise into x
compute log p(x)
store into node:  log p(x)
```

**Backward:**
```
get from message:  incoming loss with the baseline subtracted if available
get from node:  log p(x)
message:  (tensors=log p(x), grad_tensors=loss)
```

---

## LRProp

This propagator adds optional backpropagation and baseline functionality to the base propagator for LR gradients.

**Sequence:** `[Optional(BackPropagator), LRBase, Optional(BaselineProp)]`

Notice that in `LRProp` the `BaselineProp` is at the right side of the sequence, whereas for `LossProp` it is at the left side. The backward sequence commences from right to left. In the `LRProp` case, we want to subtract the baseline before estimating the LR gradient, whereas in the `LossProp` case, we want to subtract the baseline in the end, after having computed the loss. Which one is better is problem dependent—if there is a single loss node, but many LR nodes, it may be better to subtract the baseline at the loss node, and *vice versa*.

## TPBase

This is the base propagator for the most complicated propagator we show in our examples here—it implements the computations for the total propagation algorithm.

**Configuration:**   what node to use for inverse variance weighting

**Forward:**
```
input from program code:  x
inject reparameterized noise into x
compute log p(x)
store into node:  log p(x), output, detached output
```

**Backward:**
```
get from message:  incoming loss with the baseline subtracted if available
get from node:  log p(x), output, detached output
get grads from detached output
compute LR and RP gradients until the inverse gradient variance node
perform inverse variance weighting and compute the mixing ratio, k
message:  (tensors=[log p(x), x], grad_tensors=[k*loss, (1-k)*grads])
```

Note that here we are packing two sets of `tensors` and `grad_tensors` together into a list. The `torch.autograd.backward()` function that is called in `BackPropagator` can handle lists of such pairs, and simultaneously invoke the backpropagtion, so there is no issue.

## TotalProp

Finally, this propagator adds the back propagation and baseline functionalities to the base propagator for TP gradients.

**Sequence:** `[Optional(BackPropagator), TPBase, Optional(BaselineProp)]`

### B.3   Computational time comparison

**Setup.**   To give an indication of the overhead in computational time caused by using Proppo, we perform experiments with a similar recurrent neural network as in Sec. 4.1, but while varying the batch size (Fig. 5) or dimensionality (Fig. 6). When varying the batch size, the dimensionality was fixed to 500; and when varying the dimensionality, the batch size was fixed to 1000. The horizon was 10. We performed the forward and backward computations 100 times and estimated the average computation time both on a CPU and on a GPU. The computation times were normalized with the minimum computation time at the corresponding setting to better highlight the ratio difference. We compared the computation times of RP, LR, TP and when implementing RP without using Proppo. Moreover, for RP, we test two implementations: one which detaches the tensors at each propagation node, and manually back propagates the gradients; and another that does not detach the tensors, and allows PyTorch to handle the gradient back propagation.

**Discussion.**    In the results in Figs. 5 and 6 we see that for very small problem sizes, Proppo causes a significant overhead, but as the problem size becomes larger, the overhead becomes negligible. For large problem sizes, typically TP required 2 times more computational time. Compared to a 100 time reduction in gradient variance (Sec. 4.1), this additional computational time is negligible (note that the naïve way to reduce gradient variance by a factor $K$ is to increase the batch size by the same factor $K$ roughly requiring $K$ times more computational time). We also note that in a typical full implementation of an ML algorithm, Proppo may be used in only some section of the computations. In this case, the overhead caused by Proppo may be only a small fraction of the total computational time associated with the algorithm. We observed this point in our concurrent work in model-based reinforcement learning (Anonymous, 2022), where the total change in computational time was typically less than 50% extra.

Another point to note is the difference between the used *framework* and *implementation* of an algorithm. The performance will primarily depend on the implementation. Proppo allows creating multiple implementations, fast or slow ones (e.g., compare the two implementations of RP). The main point is not so much that the computation time might increase a bit if Proppo is used, but rather that using Proppo enables implementing algorithms that would be cumbersome to implement otherwise. The current implementations are also not fully tuned, and can be implemented to run faster. In particular, the implementation of TP has issues with scalability if the number of parameters used for the inverse variance weighting becomes large, but we have found sensible solutions to this issue in our concurrent work (Anonymous, 2022).

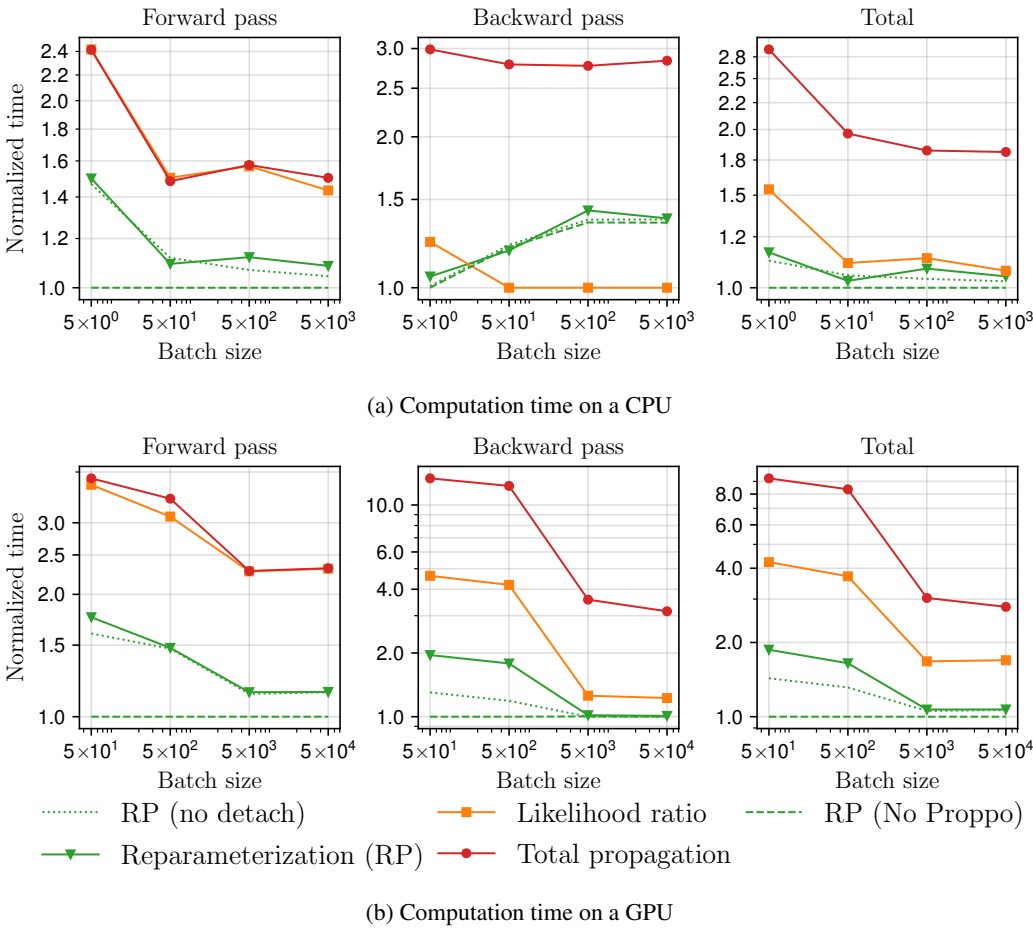

(a) Computation time on a CPU

(b) Computation time on a GPU

Figure 5: Computation times of algorithms in Proppo when varying the batch size (Sec. B.3).

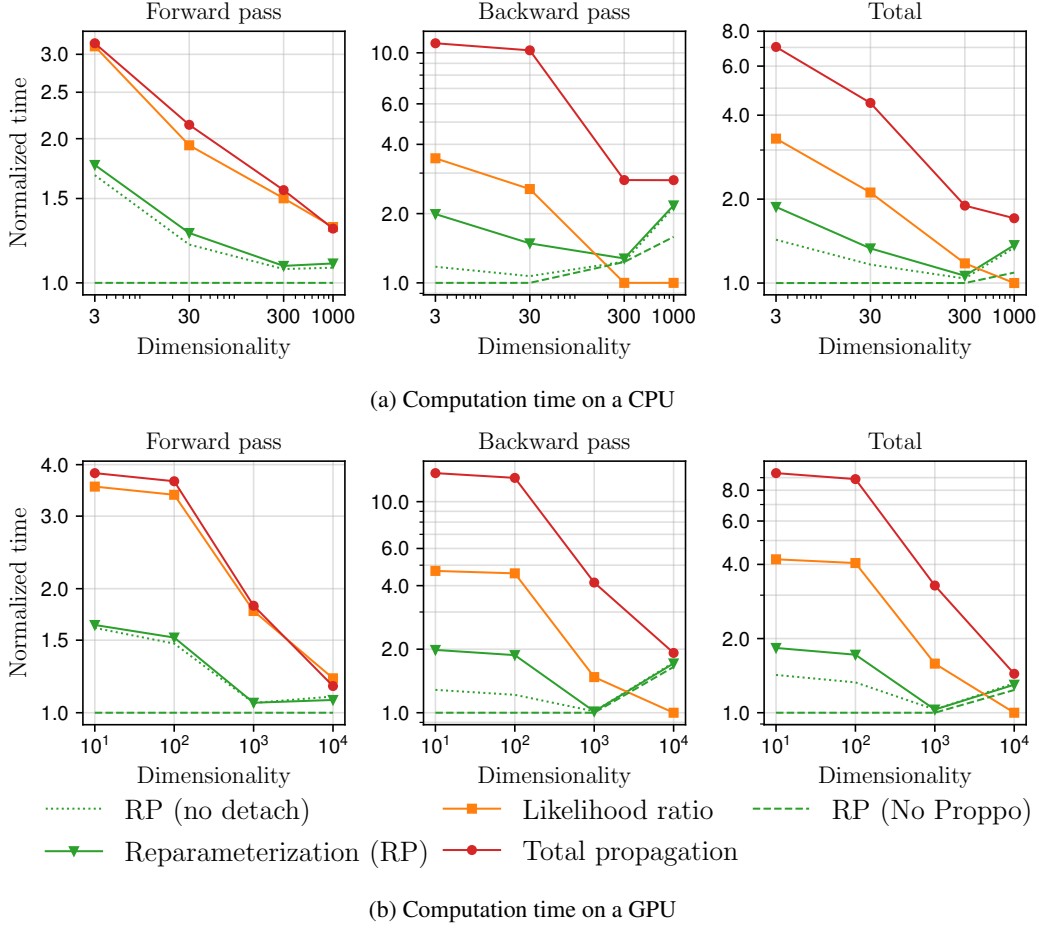

(a) Computation time on a CPU

(b) Computation time on a GPU

Figure 6: Computation times of algorithms in Proppo when varying the dimensionality (Sec. B.3).

## B.4 Scaling with the number of nodes

**Setup.** We perform an experiment to test how Proppo scales with the number of propagation nodes. In the experiment, we connect nodes into a chain with a dummy propagator that performs no computations, but merely stores an empty node in the forward pass, and sends empty messages in the backwards pass. This experiment allows testing the overhead associated with the "bookkeeping" performed by Proppo. We ran 10 experiments, and averaged the elapsed time. We tested values between 10 and $10^7$ nodes, and the experiment was performed on an Intel i9 CPU. In Fig. 7, we show the results separately for the forward and backwards pass, as well as the total elapsed time.

**Discussion.** The results in Fig. 7 show that the computation time increases linearly. In particular, we see that the time spent per node is roughly $10^{-6}$ seconds. This means that if the computations performed by the propagator take longer than $10^{-6}$ seconds, we would expect the additional overhead for storing nodes and passing messages caused by Proppo to be negligible. This overhead will be non-negligible for extremely simple calculations such as summing scalars together; however, the aim of Proppo is to facilitate implementing complicated algorithms. For complicated algorithms, we expect the computations at each node to take longer than $10^{-6}$ seconds, so in practice, the overhead caused by Proppo will be negligible. Moreover, we see that Proppo can scale to millions of propagation nodes, while, in our experience applying Proppo to practical problems, we have so far not needed more than a thousand nodes. Finally, we note that this experiment does not guarantee that the implementation utilizing Proppo will have the same computation time as one without it. Typically, when Proppo is used, it would tweak the operation of some underlying computational software. These tweaks may interfere with the normal operation of the computational software, making it perform

slightly slower. For example, when Proppo is used to override the standard back propagation in PyTorch, it would manually pass gradients backwards at the propagation nodes, instead of letting PyTorch automatically back propagate. This may slow down the computation due to the interference. To obtain optimal performance, one should only use Proppo when it is needed, or when it increases convenience due to greater modularity.

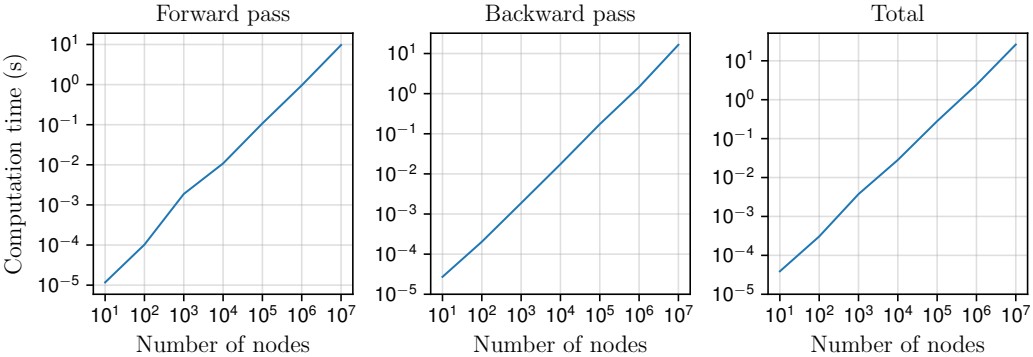

Figure 7: Scaling of computation time with the number of propagation nodes (Sec. B.4).

## C  Chaotic net: Additional details, explanations and experiments

In Sec. C.1, we provide additional explanations about the chaotic net experiments (Sec. 4); in Sec. C.2, we provide additional explanations of the single path experiment (Sec. 4.2); in Sec. C.3, we present an additional experiment designed to show the utility and better scalability w.r.t. the dimensionality of Gaussian shaping gradients.

### C.1  Explanation of chaotic dynamics of the recurrent neural network

Wang (1991) explained that the inverse temperature parameter $\beta$ controls the dynamics of the RNN system introduced in Sec. 4.1. When $\beta$ is small, the system is well-behaved, but as $\beta$ is increased, the system becomes chaotic through a period-doublings mechanism. They illustrated this via a bifurcation diagram that we have replicated in Fig. 8a. In this figure, for each $\beta$ we simulate the system for 10000 steps, and plot the first dimension $x_1$ of the last 500 steps onto the figure as dots (note that one obtains a similar diagram when considering $x_2$, the second dimension of the system). We see that when $\beta$ is small (around 1), all of the dots are at the same position; hence the system converged to a fixed state. As $\beta$ is increased, the system starts oscillating between two states. As $\beta$ is further increased, the states further split, with a phase transition happening around $\beta = 2.5$ leading to chaotic oscillation.

This chaotic behavior causes the gradient to be ill-behaved as illustrated in Fig. 8b. In this figure, we followed the experimental protocol of Parmas et al. (2018), and plotted the RP derivative of the loss function w.r.t. $\beta$ while keeping the random number seed fixed. We see that the derivative is well-behaved in the non-chaotic region, but starts rapidly oscillating up and down with a large magnitude as the system becomes chaotic at the phase transition around $\beta = 2.5$. Moreover, the gradient variance also explodes. Parmas et al. (2018) explained this behavior by plotting the loss landscape of their system w.r.t. the start position. We have replicated a similar result for the RNN system in Fig. 8c. Recall from the preliminaries (Sec. 2, see also Eq. (7)) that the total derivative sums the terms $\frac{dL}{dx}\frac{dx}{d\beta}$ for each time step. As the loss landscape $L$ has a fractal structure due to the chaotic properties of the system, the gradient $\frac{dL}{dx}$ is oscillating rapidly. Thus, if one tries to average the gradients together over some region of this landscape by sampling the gradients in said region, the variance of this estimate will explode, and it is impossible to compute a sensible gradient direction using the RP method. The LR gradient, on the other hand, does not use $\frac{dL}{dx}$, it only uses $L$ to estimate the gradient. Thus, small amplitude fluctuations of $L$ do not affect the LR gradient, and it is robust to the issues with chaos.

Similar chaotic properties occur in many ML tasks. We already introduced the work of Parmas et al. (2018) in model-based reinforcement learning that was the basis for much of our discussion. Their work appears to be the first to discuss the explosion of the gradient variance due to chaos when the computations are stochastic, and they also suggested to incorporate LR methods to tackle the issue. Similar chaotic properties have also been discussed in metalearning (?Metz et al., 2019), protein folding software (?) and differentiable simulation (??). Moreover, the effect of chaos on estimating the sensitivity of fluid simulations has also been studied in many works, e.g., the work by ?. We believe that our minimalistic experiment captures interesting characteristics of such challenging ML tasks.

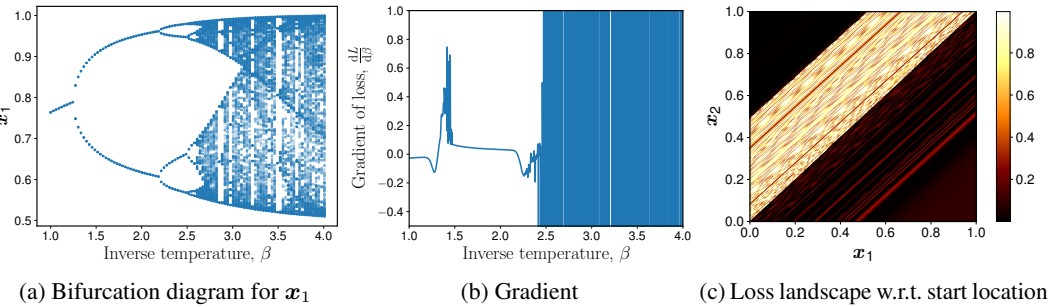

| (a) Bifurcation diagram for $x_1$ | (b) Gradient | (c) Loss landscape w.r.t. start location |

Figure 8: Illustration of the behavior of the chaotic sigmoid recurrent neural network (Sec. 4.1). (a) Bifurcation diagram of the RNN. The activation noise was removed, $\sigma = 0$, to replicate previous results with a deterministic RNN (Wang, 1991). Note that the result does not change much when noise is added, the dots are just spread out around the location on the current figure. (b) Gradient of the objective plotted against $\beta$, similarly to (Parmas et al., 2018). The horizon was $H = 100$. (c) Fractal loss landscape. The parameters were $\beta = 3.5$, resolution of the grid: $500 \times 500$, horizon: $H = 15$. We also added a fixed perturbation on $W_{11}$ sampled from a Gaussian with standard deviation $\sigma_w = 0.1$. This was done for aesthetic reasons, and to show that the system stays chaotic even when perturbed.

## C.2    Additional details of the single path chaotic net experiment

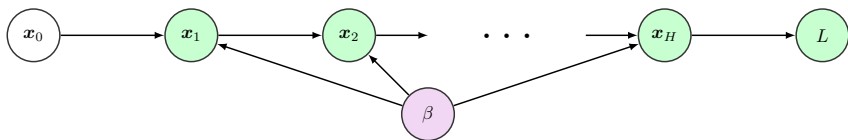

Figure 9: Single path chaotic RNN probabilistic computation graph diagram.

Figure 9 includes a schematic diagram of the RNN computations in the experiments in Sec. 4.2. The computations start from an initial state $x_0$, and are simulated forwards for $H$ steps. A loss, $L$, is computed at the last step. All of the nodes when $t \geq 1$ are propagation nodes ◯. The $x$ nodes are propagation nodes due to the sampling of noise, and gradient estimation, while the loss node, $L$, is implemented as a loss propagator. The $\beta$ node ◯ is the parameter node where the inverse variance weights are computed for use in the TP algorithm.

## C.3    Multi path experiment: showing the advantage of Gaussian shaping gradients

**Experimental setup.**    In this section, the main aim is to show the better scalability of Gaussian shaping gradients compared to regular LR gradients. To this end, we modify the simple RNN in Sec. 4.1 by replicating multiple instances of this RNN, and computing them in parallel; thus, increasing the dimensionality of the system. The setup is illustrated in Fig. 10. The additional parallel dimensions act as nuisance variables on the final loss, increasing the variance of the LR gradient estimator. Formally, the state is modified into $\tilde{x} := [x^{(1)}; \ldots; x^{(D)}]$, and the evolution of each $x^{(d)}$ is computed separately according to Eq. (2). While the initial state is the same for each $d$, the added activation noise $\varepsilon$ is different, so the trajectories for each parallel path are different. The loss is computed at the final step as $\tilde{L}(\tilde{x}_H) = \frac{1}{2}(\tilde{x}_H - \mathbf{1})^{\mathrm{T}}(\tilde{x}_H - \mathbf{1})$. Note that the parameters of the

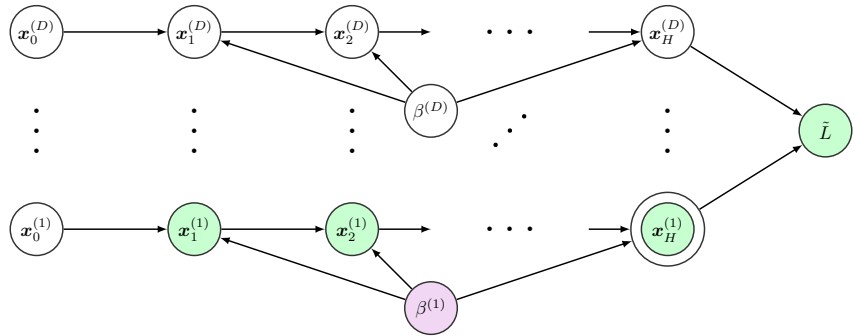

Figure 10: Multi path chaotic RNN probabilistic computation graph diagram.

networks $W$ and $\beta$ are *not* shared across the paths, and we estimate the gradient only through the first path according to $\beta^{(1)}$. If instead the $\beta$ parameter were shared, the problems of LR demonstrated in this section would not appear. Gaussian shaping modifies the gradient estimation method by resampling the batch in the first path at the last step $\{\boldsymbol{x}_H^{(1,m)}\}_{m=1}^B$ from a fitted Gaussian distribution (this node is illustrated as ◎ in Fig. 10). The other experimental details are the same as before in Sec. 4.2. We estimate the gradient when $\beta \in \{2.0, 2.5, 3.5\}$, and plot the gradient variance against the number of parallel paths $D$ (Fig. 11).

**Results.** The results are in Fig. 11. We see that the regular LR has a linearly increasing variance as the dimensionality $D$ is increased, whereas the variance of GS stays constant. The variance of RP also stays constant with the dimensinality irrespective of whether GS is used or not; however, RP is inaccurate in the chaotic regime with $\beta \in \{2.5, 3.5\}$. The variance of TP follows a similar pattern to LR; however, for $\beta = 2.0$ the variance of the regular TP does not increase, because RP gradients are accurate in that scenario. We also see that TP outperformed the other estimators in all cases.

**Discussion.** Previously, Parmas (2018) explained a potential advantage of GS in terms of the bias that it adds—smoothing the loss with a Gaussian may promote unimodality simplifying some optimization problems. However, they did not demonstrate a fundamental advantage in terms of computational complexity. Here, on the other hand, we have demonstrated a fundamental advantage of GS in terms of its scalability with the dimensionality $D$. This newly shown effect is particularly important when the system is near-chaotic, and regular back propagation gradients are ill-behaved.

Our newly shown advantage of GS may pose it beneficial for optimizing complex model architectures. When there are several modules independently influencing the behavior of downstream components of the system, GS may disentangle the individual contributions, and enable efficient optimization. As GS is not practical to implement using standard automatic differentiation software, our result highlights that automatic propagation software such as Proppo may enable training previously untrainable machine learning systems composed of complex networks of connected modules.

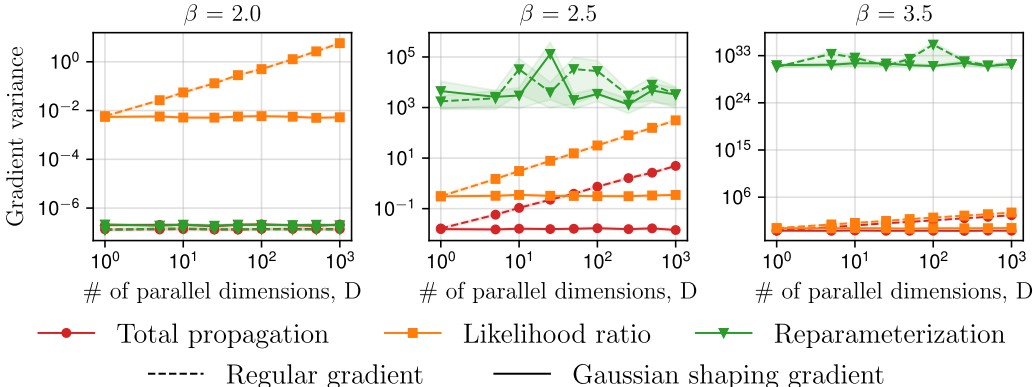

Figure 11: Comparison of Gaussian shaping (GS) gradient estimators with their regular counterparts in terms of scalability w.r.t. the dimensionality of the system $D$. The discussion of the experiment is in Sec. C.3. The main result is that the variance of the regular LR increases linearly as $D$ is increased, whereas the GS variant has a constant variance.

# D    Proppo prototype code listings

This section contains prototype code for Proppo, as well as its application to create MC gradient estimator propagators. The code describes one possible prototype of automatic propagation software, but there may also be other implementations. The code here is not intended for use—it is provided for archival purposes. Code for use is uploaded to https://github.com/proppo/proppo.

**Propagation manager.**

```python
1  # propagation_manager.py
2  import torch
3  import proppo.back_methods as back_methods
4  import proppo.propagators as propagators
5  from proppo.containers import Node, Message
6
7
8  def _reversed_enumerate(l):
9      count = len(l)
10     for value in reversed(l):
11         count -= 1
12         yield count, value
13
14
15 class PropagationManager:
16     """ Propagation Manager class.
17     This class enables custom forward and backward propagations for flexibly
18     designing new gradient estimation and learning algorithms for computational
19     graphs, e.g. neural networks.
20
21     """
22
23     def __init__(self,
24                  default_propagator=propagators.BackPropagator(),
25                  loss_propagator=None,
26                  terminal_propagator=propagators.BackPropagator()):
27         self.nodes = []
28         self.node_pointer = 0  # Pointer for the current position on the tape.
29         self.default_propagator = default_propagator
30         self.propagators = {}
31         if loss_propagator:
32             self.loss_propagator = loss_propagator
33         elif default_propagator:
34             self.loss_propagator = self.default_propagator.loss_propagator()
35         # The terminal propagator exists to handle any remaining messages
36         # once the backward pass has finished. For example, a common use
37         # case is to use BackPropagator() to call backprop once all
38         # outputs and gradients have been assembled for the backprop call,
39         # # if these can be performed in parallel.
40         self.terminal_propagator = terminal_propagator
41         if self.terminal_propagator != None:
```

```
42                    self.forward(x=None, local_propagator=terminal_propagator)
43                    self.nodes[0]['clear'] = False
44
45        def add_propagator(self, name, propagator):
46            self.propagators[name] = propagator
47
48        def forward(
49                self,
50                x,
51                force_targets=None,
52                local_propagator=None,
53                get_node=False,
54                **kwargs):
55            """ Register input as a node, and returns output through local forward
56                function.
57
58            """
59            # Do modification to x, e.g. add noise
60            if local_propagator != None:
61                if isinstance(local_propagator, str):
62                    local_propagator = self.propagators[local_propagator]
63                node = local_propagator.forward(x, **kwargs)
64            else:
65                node = self.default_propagator.forward(x, **kwargs)
66
67            if not isinstance(node, Node):
68                node = Node.from_container(node)
69
70            # set local backward propagator for later use with the
71            # propagator.backward method
72            node.assign_propagator(local_propagator)
73
74            if force_targets != None:
75                node['force_targets'] = force_targets
76
77            if 'output' in node:
78                output = node['output']
79            else:
80                output = None
81
82            if node['register_node']:
83                if self.node_pointer > (len(self.nodes) - 1):
84                    self.nodes.append(node)
85                else:
86                    self.nodes[self.node_pointer] = node
87                self.node_pointer += 1
88
89            # Return output as well as pointers to the messages and
90            # node content.
91            # The node pointer is given if one wants to retrospectively change
92            # something in the node, e.g. if one wants to target messages to
93            # future nodes.
94
95            if get_node:
96                return (output, self.nodes[self.node_pointer])
97            else:
98                return output
99
100
101        def backward(self, loss=None, clear_nodes=False, message=None):
102            """ Execute backward propagation at the registered nodes one by one.
103
104            """
105            # backward until the last node
106            if loss != None:
107                self.append_loss(loss)
108
109            if message != None:  # TODO, send message to node instead.
110                self._send_message(
111                    message=message
112                )  # message should be of Message type and include the target
113
114            # Loop through all nodes in reverse order, calling the
115            # custom backward method of that node
116            for i, node in _reversed_enumerate(self.nodes):
117                self.node_pointer = i
118                if node.propagator != None:
119                    messages = node.backward()
120                else:
121                    messages = self.default_propagator.backward(
122                        node, node.messages)
```

```
123
124             # Clear node and message tape content, then send the message;
125             # this allows to send a message to the propagators own slot
126             # as well (i.e. to keep a history between different
127             # manager.backward() calls). Node can optionally be not cleared,
128             # but message_tape is always cleared. (To keep a history, in
129             # message tape, the propagator should send a message to itself).
130             if node['clear']:
131                 self.nodes[i].clear()
132             self.nodes[i].messages.clear()
133             if messages != None:  # TODO: loop through the message items
134                 for target, message_container in messages.messages():
135                     # choose the priority target, transform the target, send the message
136                     targets = self._target_conflict_resolution(target, node)
137                     target_nodes = self._find_nodes(targets)
138                     self._send_messages(message_container, target_nodes)
139
140         # refresh node history
141         if clear_nodes:  # TODO: remove message tape
142             if self.terminal_propagator:
143                 self.nodes = [self.nodes[0]]
144             else:
145                 self.nodes = []
146
147         if self.terminal_propagator:
148             self.node_pointer = 1
149         else:
150             self.node_pointer = 0
151
152         return messages  # The last remaining messages are returned if desired.
153
154     def _send_message(self, message, target_node):
155         if target_node != None:
156             target_node.receive(message)
157
158     def _send_messages(self, messages, targets):
159         if isinstance(targets, list):
160             for target in targets:
161                 self._send_message(messages, target)
162         else:
163             self._send_message(messages, targets)
164
165     def _target_conflict_resolution(self, target, node):
166         if 'force_targets' in node:
167             targets = node['force_targets']
168             return targets
169         elif target != None:
170             return target
171         elif 'targets' in node:
172             return node['targets']
173         else:
174             targets = -1
175             return targets
176
177     def _find_node(self, target):
178         if isinstance(target, int):
179             node_index = target + self.node_pointer
180             if (node_index < 0) or (node_index > (len(self.nodes) + 1)):
181                 target = None  # If out of bounds, don't send
182             else:
183                 target = self.nodes[node_index]
184         return target
185
186     def _find_nodes(self, targets):
187         if isinstance(targets, (list, tuple)):
188             targets = [self._find_node(t) for t in targets]
189             return targets
190         else:
191             targets = self._find_node(targets)
192             return targets
193
194     def append_loss(self, loss_node, loss_propagator=None, **kwargs):
195         if loss_propagator is None:
196             loss_propagator = self.loss_propagator
197         out = self.forward(x=loss_node,
198                            targets=None,
199                            local_propagator=loss_propagator,
200                            get_message_box=False,
201                            get_node=False,
202                            **kwargs)
203         return out
```

```
204
205     def size(self):
206         """ Returns the number of the registered nodes.
207
208         Returns:
209             int: the number of nodes.
210
211         """
212         return len(self.nodes)
```

Listing 1: Example prototype Python code for a Propagation Manager class

### Smart containers and contents.

```
 1  # containers.py
 2  from typing import List
 3
 4  class Content():
 5
 6      __slots__ = ('_content', )
 7
 8      def __init__(self, content):
 9          if isinstance(content, Content):
10              self._content = content.get()
11          else:
12              self._content = content
13
14      def get(self):
15          return self._content
16
17      def set(self, value):
18          if isinstance(value, Content):
19              v = value.get()
20          else:
21              v = value
22
23          self._content = v
24
25      def update(self, value):
26          if hasattr(self.get(), 'update'):
27              if isinstance(value, Content):
28                  v = value.get()
29              else:
30                  v = value
31              self.get().update(v)
32          else:
33              self.set(value)
34
35      def __repr__(self):
36          return 'Content(' + str(self._content) + ')'
37
38      def __str__(self):
39          return str(self._content)
40
41      def __add__(self, value):
42          if isinstance(value, Content):
43              return self.get() + value.get()
44          else:
45              return self.get() + value
46
47      def __mul__(self, value):
48          return self.get() * value.get()
49
50      def __rmul__(self, value):
51          return self.get() * value.get()
52
53      def __matmul__(self, value):
54          return self.get() @ value.get()
55
56
57  class Summed(Content):
58
59      def update(self, value):
60          self.set(self + value)
61
62      def __repr__(self):
63          return 'Summed(' + str(self._content) + ')'
64
65
```

```
66   class Locked(Content):
67
68       def set(self, value):
69           raise RuntimeError(
70               'Attempting to call ".set()" on a Locked type'
71               ' Content. Locked type Content objects are used'
72               ' for contents that are never supposed to be updated.')
73
74       def __repr__(self):
75           return 'Locked(' + str(self._content) + ')'
76
77
78   class Listed(Content):
79
80       def __init__(self, *args):
81           list_content = []
82           for content in args:
83               if isinstance(content, Content):
84                   list_content.append(content.get())
85               else:
86                   list_content.append(content)
87           super().__init__(list_content)
88
89       def __repr__(self):
90           return 'Listed(' + str(self._content) + ')'
91
92       def update(self, *args):
93           if len(args) == 1:
94               if not isinstance(args[0], Listed):
95                   listed_content = Listed(args[0])
96               else:
97                   listed_content = args[0]
98           else:
99               list_vals = []
100              for arg in args:
101                  if isinstance(arg, Content):
102                      if isinstance(arg, Listed):
103                          list_vals += arg.get()
104                      else:
105                          list_vals.append(arg.get())
106                  else:
107                      list_vals.append(arg)
108              listed_content = Listed(*list_vals)
109
110          self._content = self + listed_content
111
112      def set(self, *args):
113          listed_content = []
114          for value in args:
115              if isinstance(value, Content):
116                  v = value.get()
117                  if not isinstance(value, Listed):
118                      v = [v]
119              else:
120                  v = [value]
121              listed_content += v
122
123          self._content = listed_content
124
125
126  class Container():
127
128      __slots__ = ('_contents', )
129
130      def __init__(self, cont_dict=None, **kwargs):
131          self._contents = {}
132          if cont_dict != None:
133              kwargs.update(cont_dict)
134          for k, v in kwargs.items():
135              if not isinstance(v, Content):
136                  val = Content(v)
137              else:
138                  val = v
139              self._contents[k] = val
140
141      def clear(self):
142          self._contents.clear()
143
144      def get_contents(self):
145          return self._contents
146
```

```python
147        def set_content(self, key, value):
148            if not isinstance(value, Content):
149                v = Content(value)
150            else:
151                v = value
152
153            if key in self._contents:
154                self._contents[key].set(v)
155            else:
156                self._contents[key] = v
157
158        def get(self, key):
159            return self[key]
160
161        def item_iter(self):
162            for k in self.keys():
163                yield (k, self[k])
164
165        def items(self):
166            return self.item_iter()
167
168        def keys(self):
169            return self._contents.keys()
170
171        def _update_keys(self):
172            return self._contents.keys()
173
174        def value_iter(self):
175            for k in self.keys():
176                yield self[k]
177
178        def values(self):
179            return self.value_iter()
180
181        def pop(self, key):
182            return self._contents.pop(key).get()
183
184        def __str__(self):
185            return 'Contents: ' + str(self._contents)
186
187        def __iter__(self):
188            return iter(self.keys())
189
190        def __getitem__(self, k):
191            return self._contents[k].get()
192
193        def __setitem__(self, k, v):
194            self.set_content(k, v)
195
196        def __contains__(self, k):
197            return k in self._contents
198
199        def update(self, container):
200            # Works for both dict or container.
201            if isinstance(container, dict):
202                c = Container(cont_dict=container)
203            else:
204                c = container
205
206            for k in c._update_keys():
207                if k in self._update_keys():
208                    self._contents[k].update(c._contents[k])
209                else:
210                    self.set_content(k, c._contents[k])
211
212
213    class Node(Container):
214        __slots__ = ('_contents', 'messages', 'propagator')
215
216        def __init__(self,
217                     cont_dict=None,
218                     box_class=Container,
219                     propagator=None,
220                     **kwargs):
221            super().__init__(cont_dict=cont_dict, **kwargs)
222            self.messages = box_class()
223            self.propagator = propagator
224
225        @classmethod
226        def from_container(cls, container):
227            kwargs = {}
```

```python
228             if 'box_class' in container:
229                 kwargs['box_class'] = container.pop('box_class')
230             if 'propagator' in container:
231                 kwargs['propagator'] = container.pop('propagator')
232             if isinstance(container, Container):
233                 cont_dict = container.get_contents()
234             elif isinstance(container, dict):
235                 cont_dict = container
236             else:
237                 raise TypeError('container must be of Container or dict type.')
238
239             return Node(cont_dict=cont_dict, **kwargs)
240
241         def forward(self, x, **kwargs):
242             return self.propagator.forward(x, **kwargs)
243
244         def backward(self):
245             return self.propagator.backward(self, self.messages)
246
247         def receive(self, message):
248             # TODO: add more message box classes
249             if isinstance(message, Message):
250                 for m in message.containers():
251                     self.messages.update(m)
252             else:
253                 self.messages.update(message)
254
255         def assign_propagator(self, propagator):
256             self.propagator = propagator
257
258
259 class Message(Container):
260     __slots__ = ('_contents', 'multi_message')
261
262     def __init__(self, cont_dict=None, target=-1, container=None, **kwargs):
263         if container == None:
264             super().__init__(
265                 cont_dict={target: Container(cont_dict=cont_dict, **kwargs)})
266         else:
267             super().__init__(cont_dict={target: container})
268         self.multi_message = False
269
270     def _switch_multi(self):
271         if len(self.targets()) > 1:
272             self.multi_message = True
273         else:
274             self.multi_message = False
275
276     def _get_main_message(self):
277         return self._contents[next(iter(self.targets()))].get()
278
279     def get_message(self, target):
280         return self._contents[target].get()
281
282     def __str__(self):
283         return 'Message: ' + str(self._contents)
284
285     def targets(self):
286         return Container.keys(self)
287
288     def iter_containers(self):
289         for t in self.targets():
290             yield self.get_message(t)
291
292     def containers(self):
293         return self.iter_containers()
294
295     def iter_messages(self):
296         for t in self.targets():
297             yield (t, self.get_message(t))
298
299     def messages(self):
300         return self.iter_messages()
301
302     def iter_keys(self):
303         for container in self.containers():
304             for k in container.keys():
305                 yield k
306
307     def keys(self):
308         if self.multi_message == False:
```

```python
309                return self._get_main_message().keys()
310            else:
311                return self.iter_keys()
312
313    def iter_items(self):
314        for container in self.containers():
315            for item in container.items():
316                yield item
317
318    def items(self):
319        if self.multi_message == False:
320            return super().items()
321        else:
322            return self.iter_items()
323
324    def iter_values(self):
325        for container in self.containers():
326            for value in container.values():
327                yield value
328
329    def values(self):
330        if self.multi_message == False:
331            return super().values()
332        else:
333            return self.iter_values()
334
335    def pop(self, k):
336        if self.multi_message == True:
337            out = super().pop(k)
338            self._switch_multi()
339            return out
340        else:
341            return self._get_main_message().pop(k)
342
343    def pop_message(self, k):
344        out = super().pop(k)
345        self._switch_multi()
346        return out
347
348    def update(self, message):
349        if isinstance(message, Message):
350            super().update(message)
351            self._switch_multi(
352            )  # check whether a message with a new target was added
353        elif isinstance(message, (Container, dict)):
354            if self.multi_message == False:
355                self._get_main_message().update(message)
356            else:
357                raise TypeError(
358                    'Current Message contains multiple targets. Updating'
359                    ' with Container is disabled due to ambiguity'
360                    ' in the target. Turn container into Message'
361                    ' type with a specified target, then update'
362                    ' the current message.')
363        else:
364            raise TypeError('Message can only be updated with a Message'
365                            ' or Container type.')
366
367    def __getitem__(self, k):
368        if self.multi_message == False:
369            return self._get_main_message()[k]
370        else:
371            return super().__getitem__(k)
372
373    def __setitem__(self, k, v):
374        if self.multi_message == False:
375            self._get_main_message().set_content(k, v)
376        else:
377            super().__setitem__(k, v)
378
379    def __contains__(self, k):
380        if self.multi_message == False:
381            return k in self._get_main_message()
382        return super().__contains(k)
```

Listing 2: Example prototype Python code for smart contents and containers

**Smart initializers.**

```python
1  # initializers.py
2  from collections import OrderedDict
3  from functools import partial
4  import copy
5
6
7  class Init():
8
9      def __init__(self, cls, *args, defaults=None, kwdefaults=None, **kwargs):
10         self._cls = cls
11         if kwdefaults != None:
12             self.kwdefaults = {**kwdefaults, **kwargs}
13         elif kwargs != {}:
14             self.kwdefaults = kwargs
15         if defaults != None:
16             self.defaults = (*args, *defaults)
17         elif args != ():
18             self.defaults = args
19
20     def __call__(self, *args, **kwargs):
21         if hasattr(self, 'kwdefaults'):
22             input_kwargs = {**self.kwdefaults, **kwargs}
23         else:
24             input_kwargs = kwargs
25         # Note that concatenating defaults and args is not allowed
26         # due to ambiguity of when to overwrite.
27         # TODO: fix the default arguments so that they are first
28         # converted into keyword arguments, and then merged with the
29         # keyword arguments. If there is a conflict, e.g., the same
30         # keyword exists in both sets, then raise an error.
31         if hasattr(self, 'defaults') and len(args) == 0:
32             input_args = self.defaults
33         else:
34             input_args = args
35
36         return self._call(*input_args, **input_kwargs)
37
38     def _call(self, *args, **kwargs):
39         return self._cls(*args, **kwargs)
40
41     @classmethod
42     def init(cls, *args, **kwargs):
43         return Init(cls, *args, **kwargs)
44
45  class Lock(Init):
46
47     def __init__(self, *args, allow_unused=False, **kwargs):
48         super().__init__(*args, **kwargs)
49         self.allow_unused = allow_unused
50
51     def __call__(self, *args, allow_unused=False, **kwargs):
52         if allow_unused or self.allow_unused:
53             return super().__call__()
54         elif len(args) == 0 and len(kwargs) == 0:
55             return super().__call__()
56         else:
57             raise RuntimeError('allow_unused permission is False.'
58                                ' The Lock initalizer allows only'
59                                ' initializing with the default parameters.'
60                                ' Remove args and kwargs inputs from'
61                                ' initialization.')
62
63  class Optional(Init):
64     """ Initializer that only initializes when the on flag is
65     True. Otherwise, an instance of the class is not created, it returns None.
66     """
67     def _call(self, on, *args, **kwargs):
68         if on:
69             return super()._call(*args, **kwargs)
70         else:
71             return None
72
73  class Choice(Init):
74     def __init__(self, cls, *args, **kwargs):
75         if not isinstance(cls, (tuple, dict)):
76             raise TypeError('cls input must be of type tuple or dict.')
77         else:
78             super().__init__(cls, *args, **kwargs)
79
80     def _call(self, choice, *args, **kwargs):
81         return self._cls[choice](*args, **kwargs)
```

```python
82
83  class Empty(Init):
84      def __init__(self):
85          pass
86
87      def __call__(self, *args, **kwargs):
88          self._call()
89
90      def _call(self, *args, **kwargs):
91          raise RuntimeError('Trying to initialize a ChainInit containing'
92                             ' an empty Init. First reconfigure the'
93                             ' ChainInit to replace the Empty Init.'
94                             ' Empty Inits are used in template'
95                             ' ChainInits to indicate what slot has to be'
96                             ' changed.')
97
98  class ChainInit():
99
100     def __init__(self, regular_dict=False, **kwargs):
101
102         if regular_dict:
103             self._chaininit = {}
104         else:
105             self._chaininit = OrderedDict()
106
107         for k, v in kwargs.items():
108             if isinstance(v, Init):
109                 init = v
110             else:
111                 init = Init(v)
112             self._chaininit[k] = init
113
114     def __call__(self, **kwargs):
115         return self.init(**kwargs)
116
117     def items(self):
118         return self._chaininit.items()
119
120     def values(self):
121         return self._chaininit.values()
122
123     def keys(self):
124         return self._chaininit.keys()
125
126     def __iter__(self):
127         return self._chaininit.__iter__()
128
129     def __contains__(self, key):
130         return key in self._chaininit
131
132     def __getitem__(self, key):
133         return self._chaininit[key]
134
135     def __len__(self):
136         return len(self._chaininit)
137
138     @staticmethod
139     def _init_inputs(init, inputs=None):
140
141         if inputs == None:
142             obj = init()
143         elif isinstance(inputs, tuple):
144             arginputs = []
145             kwinputs = {}
146             for inp in inputs:
147                 if isinstance(inp, dict):
148                     kwinputs.update(inp)
149                 else:
150                     arginputs.append(inp)
151             obj = init(*arginputs, **kwinputs)
152         elif isinstance(inputs, dict):
153             obj = init(**inputs)
154         else:
155             obj = init(inputs)
156         return obj
157
158     def init(self, dictionary=False, **kwargs):
159
160         chainobjs = []
161         chainkeys = []
162
```

```
163            # Check that all keys exist
164            for key in kwargs:
165                if key not in self._chaininit:
166                    raise KeyError(key, 'not included in keys of the ChainInit.')
167
168            for k, init in self._chaininit.items():
169                inputs = kwargs.get(k, None) # kwarg if exists, otherwise None
170
171                obj = self._init_inputs(init, inputs)
172
173                if obj != None:
174                    chainobjs.append(obj)
175                    chainkeys.append(k)
176            if not dictionary:
177                return chainobjs
178            else:
179                return {k: obj for k, obj in zip(chainkeys, chainobjs)}
180
181        def reconf(self, **kwargs):
182            # Create a new ChainInit by reconfiguring the current
183            # ChainInit. For example, replace the Empty initializer
184            # of the current ChainInit to create a functioning ChainInit.
185            # You may also change the default parameters of the Inits.
186
187            chaindict = copy.copy(self._chaininit)
188            for k, v in kwargs.items():
189                if k not in chaindict:
190                    raise KeyError('Key does not exist in the ChainInit'
191                                   ' that you are trying to reconfigure.')
192                else:
193                    if isinstance(v, Init):
194                        chaindict[k] = v
195                    else:
196                        initializer = type(chaindict[k])
197                        cls = chaindict[k]._cls
198                        initializer = partial(initializer, cls)
199                        chaindict[k] = self._init_inputs(initializer, v)
200
201            return ChainInit(**chaindict)
202
203
204 class ChainInitTemplate(ChainInit):
205
206        def __call__(self, **kwargs):
207            initializers = super().__call__(dictionary=True, **kwargs)
208            return ChainInit(**initializers)
```

Listing 3: Example prototype python code for smart initializers in Proppo

**Propagators.**

```
1  # propagators.py
2  import proppo.forward_methods as fm
3  import proppo.back_methods as bm
4  import proppo.baseline_funcs as baselines
5  import proppo as pp
6  from proppo.utils import inverse_variance_weighting
7  from proppo.containers import Node, Message, Container
8  from proppo.initializers import (ChainInit, Optional, Init, Empty,
9                                    ChainInitTemplate)
10
11 import copy
12
13
14 class Propagator:
15     """ This pairs together the forward and backward methods.
16
17     """
18
19     def __init_subclass__(cls, **kwargs):
20         cls.default_init_kwargs = kwargs
21
22     def __init__(self, **kwargs):
23         self.default_forward_kwargs = kwargs
24
25     def forward(self, x, **kwargs):
26         # Overwrite default arguments, then pass as input
27         input_kwargs = {**self.default_forward_kwargs, **kwargs}
28
```

```
29          # Must create a new node, and pass this to forward, otherwise
30          # the propagators at different forward steps will overwrite, the
31          # contents of the previous propagation.
32          node = {}
33          node = self.forward_impl(x, node, **input_kwargs)
34
35          # Flag to store the node in manager.
36          if 'register_node' not in node:
37              node['register_node'] = True
38          # Flag to clear node in manager after backwarding the node.
39          if 'clear' not in node:
40              node['clear'] = True
41
42          if isinstance(node, dict):
43              node = Container(cont_dict=node)
44
45          return node
46
47      def forward_impl(self, x, node={}, **kwargs):
48          return node
49
50      def backward(self, node, message):
51          message_in = message
52          message_out = self.backward_impl(node, message_in)
53
54          # for backwards compatibility, convert dictionaries
55          if not isinstance(message_out, Message):
56              if isinstance(message_out, dict):
57                  if 'targets' in message_out:
58                      target = message_out.pop('targets')
59                      message_out = Message(cont_dict=message_out, target=target)
60                  else:
61                      message_out = Message(cont_dict=message_out)
62              elif isinstance(message_out, Container):
63                  message_out = Message(container=message_out)
64
65          return message_out
66
67      def backward_impl(self, node, message):
68          message_out = Message(cont_dict=message.get_contents())
69          return message_out
70
71      def loss_propagator(self):
72          """ Returns the default loss propagator that should
73          be applied when appending a loss after having called
74          manager.forward using the current propagator.
75
76          """
77          return LossProp()
78
79
80  class SequenceProp(Propagator):
81      """ Base class for sequence based propagators, used to construct them.
82
83      """
84
85      def __init_subclass__(cls,
86                            propagators=ChainInit(),
87                            **kwargs):
88          super().__init_subclass__(**kwargs)
89          cls.propagators = propagators
90
91      def _split_prop_kwargs(self, kwargs):
92          prop_kwargs = {}
93          for k in self.propagators:
94              if k in kwargs:
95                  prop_kwargs[k] = kwargs.pop(k)
96          return prop_kwargs, kwargs
97
98      def __init__(self,
99                   propagators=[], # A list of already initialized propagators.
100                  **kwargs):
101
102          default_init_kwargs = copy.copy(self.default_init_kwargs)
103
104          if propagators != []:
105              input_kwargs = {**default_init_kwargs, **kwargs}
106
107              super().__init__(**input_kwargs)
108              self.propagators = propagators
109          else:
```

```
110            def_prop_kwargs, def_init_kwargs = self._split_prop_kwargs(
        default_init_kwargs)
111            prop_kwargs, init_kwargs = self._split_prop_kwargs(kwargs)
112            input_kwargs = {**def_init_kwargs, **init_kwargs}
113
114            super().__init__(**input_kwargs)
115
116            input_prop_kwargs = {**def_prop_kwargs, **prop_kwargs}
117
118            self.propagators = self.propagators(**input_prop_kwargs)
119
120
121 class ComboProp(SequenceProp):
122     """ Combines propagators, and applies them in a sequence, updating
123     message and node in-place.
124
125     """
126
127     def forward_impl(self, x, node={}, **kwargs):
128         for prop in self.propagators:
129             node_out = prop.forward_impl(x, node, **kwargs)
130             node.update(node_out)
131         return node
132
133     def backward_impl(self, node, message):
134
135         final_message = Message(cont_dict=message.get_contents())
136         for prop in reversed(self.propagators):
137             message_in = final_message._get_main_message()
138
139             message_out = prop.backward_impl(node, message_in)
140
141             # for backwards compatibility, convert dictionaries
142             if isinstance(message_out, dict):
143                 if 'targets' in message_out:
144                     target = message_out.pop('targets')
145                     message_out = Message(cont_dict=message_out, target=target)
146                 else:
147                     message_out = Message(cont_dict=message_out)
148
149             final_message.update(message_out)
150
151         return final_message
152
153
154 class BackPropagator(Propagator):
155     """ Base propagator that will backprop gradient messages, if they
156     are sent into this propagator.
157
158     """
159
160     def backward_impl(self, node, message):
161
162         message_out = bm.backward(node, message)
163         return message_out
164
165
166 class BaselineProp(Propagator):
167     """ Class for adding a baseline subtraction to the local losses.
168
169     """
170
171     def __init__(self, baseline_func=baselines.mean_baseline, **kwargs):
172         super().__init__(**kwargs)
173         self.baseline_func = baseline_func  # Default baseline function
174
175         # Note: if one wants to change the baseline function for just one
176         # forward call compared to the default baseline function in a chain
177         # of forward propagations, then they should define a new propagator
178         # object for that new forward call. I could also allow giving an
179         # additional argument in the forward call to specify a baseline
180         # for just that node; however, this would not
181         # give a key error if someone accidentally mistypes the key, and
182         # may lead to bugs, so I avoid it.
183
184     def backward_impl(self, node, message):
185         local_loss = message.pop('local_loss')
186         # Need to remove local_loss from previous message, and create
187         # a new message to avoid duplicating loss in the ComboProp
188         # backward_impl method.
189         if isinstance(self.baseline_func, (list, tuple)):
```

```
190               baselined_loss = copy.copy(local_loss)
191               for func in reversed(self.baseline_func):
192                   baselined_loss = func(baselined_loss, node)
193           else:
194               baselined_loss = self.baseline_func(local_loss, node)
195
196           message_out = {
197               'baselined_loss': baselined_loss,
198               'local_loss': local_loss
199           }
200           return message_out
201
202
203 mcgrad_temp = ChainInitTemplate(backprop=Optional(
204     Optional.init(BackPropagator, True), False),
205                                 base=Init,
206                                 baseline=Optional(
207     Optional.init(BaselineProp, True), False)
208 )
209
210
211 class PauseBase(Propagator):
212     """ A propagator that pauses all incoming gradients, then sends the
213     combined gradient backwards.
214
215     """
216
217     def forward_impl(self, x, node={}, **kwargs):
218         node = fm.detached_output(x, **kwargs)
219         return node
220
221     def backward_impl(self, node, message):
222         message_out = bm.rp_gradient(node, message)
223         return message_out
224
225
226 class SkipProp(Propagator):
227     """ A propagator that sends all incoming messages backward
228     a determined length, skipping the nodes inbetween.
229
230     """
231
232     def __init__(self, skip=1, **kwargs):
233         super().__init__(**kwargs)
234         self.skip = skip
235
236     def forward_impl(self, x, node={}, **kwargs):
237         node['output'] = x
238         return node
239
240     def backward_impl(self, node, message):
241         message['targets'] = -self.skip
242         return message
243
244
245 class PauseProp(ComboProp, propagators=mcgrad_temp(backprop=True,
246                                                    base=PauseBase)):
247     pass
248
249
250 class SumBase(Propagator):
251     """ Propagator that adds a local variable with a different variable in
252     messages. This is usually used to accumulate a sum of variables during the
253     backward pass, e.g. sum the rewards to obtain the return in reinforcement learning.
254
255     """
256
257     def __init__(self, sum_name, local_variable, **kwargs):
258         self.sum_name = sum_name
259         self.local_variable = local_variable
260         super().__init__(**kwargs)
261
262     def backward_impl(self, node, message):
263         current_sum = message.pop(self.sum_name, 0)
264         message = {self.sum_name: message[self.local_variable] + current_sum}
265
266         return message
267
268
269 class ChainProp(SequenceProp):
270     """ Chains together a set of propagators into a single propagator.
```

```python
        The implementation is based on creating a new PropagationManager object
        to correctly apply the propagators in sequence without any implementation
        errors. The propagators to chain together should be given as a list
        or tuple of Propagator instances during creation. The propagators
        themselves can also be Chain propagators, which allows for defining
        complex propagation strategies using nested propagation managers.

        """

    def forward_impl(self, x, node={}, chain_kwargs=[], **kwargs):
        manager = pp.PropagationManager(default_propagator=None,
                                        terminal_propagator=None)
        if chain_kwargs:
            for prop, kwarg in zip(self.propagators, chain_kwargs):
                kwarg.update(kwargs)
                x = manager.forward(x, local_propagator=prop, **kwarg)
        else:
            for prop in self.propagators:
                x = manager.forward(x, local_propagator=prop, **kwargs)

        node = {'output': x, 'manager': manager}
        if manager.size() == 0:
            node['register_node'] = False
        return node

    def backward_impl(self, node, message):
        message_out = node['manager'].backward(message=message)
        return message_out

    def loss_propagator(self):
        """ By default, usually the last one in the chain
        contains the correct loss propagator.

        """
        return self.propagators[-1].loss_propagator()

class RPBase(Propagator):
    """ Base class for RP propagator.

    """

    def forward_impl(self, x, node={}, detach=True, **kwargs):
        node = fm.rp_noise(x, detach=detach, **kwargs)
        return node

    def backward_impl(self, node, message):
        message_out = bm.rp_gradient(node, message)
        return message_out

class RPProp(ComboProp, propagators=mcgrad_temp(backprop=True,
                                                base=RPBase)):
    """ RP propagator combining the functionality from ComboProp.

    """
    pass

class LossBase(Propagator):
    """ Base class for loss nodes in the computational graph.

    """

    def __init__(self, loss_name='local_loss', **kwargs):
        super().__init__(**kwargs)
        self.loss_name = loss_name

    def forward_impl(self, x, node={}, lossgrad=True, lossfunc=None, **kwargs):
        if lossfunc:
            if isinstance(x, dict):
                losses = lossfunc(**x)
            else:
                losses = lossfunc(x)
        else:
            losses = x

        node = fm.loss_forward(losses, sum_loss=True, lossgrad=lossgrad)
        return node

    def backward_impl(self, node, message):
```

```
352            message_out = bm.loss_backward(node, message, loss_name=self.loss_name)
353            return message_out
354
355
356    class LossProp(ComboProp,
357                    propagators=ChainInit(
358                        backprop=Optional(BackPropagator, True),
359                        baseline=Optional(BaselineProp,
360                                          True,
361                                          baseline_func=baselines.mean_baseline),
362                    base=LossBase)):
363        """ Propagator adding Baseline and ComboProp functionality to Loss nodes.
364
365        """
366        pass
367
368
369    class LRBase(Propagator):
370        """ Base class for likelihood ratio gradient propagators.
371
372        """
373
374        def forward_impl(self, x, node, **kwargs):
375            node = fm.lr_noise(x, **kwargs)
376            return node
377
378        def backward_impl(self, node, message):
379            message_out = bm.lr_gradient(node, message)
380            return message_out
381
382
383    class LRProp(ComboProp, propagators=mcgrad_temp(backprop=True,
384                                                   base=LRBase,
385                                                   baseline=True)):
386        """ Class adding ComboProp functionality to LR gradient propagators.
387
388        """
389
390        def loss_propagator(self):
391            return LossProp(backprop=False, lossgrad=False)
392
393
394    class TPBase(Propagator):
395        """ Base class for total propagation gradient propagators.
396
397        """
398
399        def __init__(self,
400                     var_weighting_func=inverse_variance_weighting,
401                     **kwargs):
402            super().__init__(**kwargs)
403            self.var_weighting_func = var_weighting_func
404
405        def forward_impl(self, x, node, **kwargs):
406            node = fm.totalprop_noise(x, **kwargs)
407            return node
408
409        def backward_impl(self, node, message):
410            message_out = bm.totalprop_gradient(
411                node, message, var_weighting_func=self.var_weighting_func)
412            return message_out
413
414
415    class TotalProp(ComboProp,
416                    propagators=mcgrad_temp(backprop=True,
417                                            base=TPBase,
418                                            baseline=True)
419    ):
420        """ Class adding ComboProp functionality to total propagation
421        gradient propagation nodes.
422
423        """
424        pass
```

Listing 4: Example prototype python code for propagators for MC gradient estimation

**Forward methods.**

```
1    # forward_methods.py
```

```python
import torch
import collections

def detached_output(x, requires_grad=True, **kwargs):
    x_detached = x.detach()
    if requires_grad:
        x_detached.requires_grad_()
    node = {'output': x_detached, 'pre_output': x, 'register_node': True}
    return node

def loss_forward(x, lossgrad=False, sum_loss=True, **kwargs):
    """ Method to register a loss node. It will either
    just pass the loss in the local_loss slot, or will
    also backprop the gradient.

    """
    if x.dim() == 1:
        x = x.reshape([x.numel(), 1])

    node = detached_output(x, requires_grad=False)
    node['lossgrad'] = lossgrad
    node['sum_loss'] = sum_loss
    return node

def rp_noise(x, dist_class, dist_params, detach=True, **kwargs):
    """ Returns noisy node for reparametrization trick

    """
    if isinstance(dist_params, (tuple, list)):
        dist = dist_class(*dist_params)
    if isinstance(dist_params, dict):
        dist = dist_class(**dist_params)
    if isinstance(dist_params, collections.Callable):
        dist = dist_class(**dist_params(x))

    x_noisy = x + dist.rsample()

    if detach:
        node = detached_output(x_noisy)
    else:
        node = {'output': x_noisy, 'register_node': False}
    return node

def lr_noise(x, dist_class, dist_params, requires_grad=False, **kwargs):
    """ Returns noisy node for likelihood ratio

    """
    if isinstance(dist_params, (tuple, list)):
        dist = dist_class(*dist_params)
    if isinstance(dist_params, dict):
        dist = dist_class(**dist_params)
    if isinstance(dist_params, collections.Callable):
        dist = dist_class(**dist_params(x))

    x_noisy = x + dist.rsample()
    log_prob = dist.log_prob(x_noisy.detach() - x)
    node = detached_output(x_noisy, requires_grad=requires_grad)
    node['log_prob'] = log_prob
    return node

def totalprop_noise(x,
                    dist_class,
                    dist_params,
                    ivw_target,
                    k_interval=1,
                    **kwargs):
    """ Returns noisy node for total propagation

    """
    node = lr_noise(x,
                    dist_class=dist_class,
                    dist_params=dist_params,
                    requires_grad=True)
    node['ivw_target'] = ivw_target
    node['k_interval'] = k_interval
```

```
82        return node
```

Listing 5: Example prototype python code for the forward methods of the propagators

**Backward methods.**

```python
 1  # back_methods.py
 2  import torch
 3  from proppo.utils import inverse_variance_weighting
 4  from proppo.containers import Listed
 5
 6
 7  def backward(node, message, grad_name=None):
 8      """ This one is special for calling backward.
 9      Removes outputs and grads from message, calls backward,
10      and passes the remaining message backwards.
11
12      """
13      if 'outputs' in message:
14          outputs = message.pop('outputs')
15          grads = message.pop('grads')
16      else:
17          return message
18
19      torch.autograd.backward(tensors=outputs, grad_tensors=grads)
20      return {}
21
22
23  def loss_backward(node, message_in, loss_name):
24      local_loss = node['output']
25      if node['sum_loss']:
26          if loss_name in message_in:
27              local_loss += message_in[loss_name]
28
29      outputs = node['pre_output']
30      # Grads is set so that the gradient of the average loss is computed.
31      message = {loss_name: local_loss}
32
33      if node['lossgrad']:
34          ones_matrix = torch.tensor([1.0], device=outputs.device).expand(
35              outputs.size())
36          grads = {
37              'outputs': Listed(outputs),
38              'grads': Listed(ones_matrix / torch.numel(outputs))
39          }
40          message.update(grads)
41
42      return message
43
44
45  def rp_gradient(node, message_in):
46      """ Returns output tensor and its gradient for reparametrization trick.
47
48      """
49      detached_output = node['output']
50      output = node['pre_output']
51      message = {
52          'outputs': Listed(output),
53          'grads': Listed(detached_output.grad)
54      }
55
56      return message
57
58
59  def lr_gradient(node, message_in):
60      """ Returns output tensor and its gradient for likelihood ratio
61
62      """
63      if 'baselined_loss' in message_in:
64          local_loss = message_in['baselined_loss']
65      else:
66          local_loss = message_in['local_loss']
67
68      lr_grad_outputs = local_loss / torch.numel(local_loss)
69      log_prob = node['log_prob']
70
71      lr_grad_outputs = lr_grad_outputs.expand(log_prob.shape)
72
73      message = {'outputs': Listed(log_prob), 'grads': Listed(lr_grad_outputs)}
```

```
74        return message
75
76
77  def totalprop_gradient(node,
78                         message_in,
79                         var_weighting_func=inverse_variance_weighting):
80      """ Returns output tensors and their gradients for total propagation.
81
82      The total propagation is a combination of the reparametrization trick and
83      the likelihood ratio.
84      Each gradient will be combined based on inverse variance weighting.
85
86      """
87      detached_output = node['output']
88      output = node['pre_output']
89      log_prob = node['log_prob']
90      ivw_target = node['ivw_target']
91      k_interval = node['k_interval']
92
93      if 'k_counter' in message_in:
94          k_counter = message_in.pop('k_counter')
95      else:
96          k_counter = 0
97
98      if 'baselined_loss' in message_in:
99          local_loss = message_in['baselined_loss']
100     else:
101         local_loss = message_in['local_loss']
102     # Make LR gradients compute mean gradient
103     local_loss = local_loss / torch.numel(local_loss)
104
105     lr_grad_outputs = local_loss.expand(log_prob.shape)
106
107     if k_counter % k_interval == 0:
108         rp_grads = torch.autograd.grad(outputs=output,
109                                        inputs=ivw_target,
110                                        grad_outputs=detached_output.grad,
111                                        retain_graph=True)
112
113         lr_grads = torch.autograd.grad(outputs=log_prob,
114                                        inputs=ivw_target,
115                                        grad_outputs=lr_grad_outputs,
116                                        retain_graph=True)
117
118         k_lr, k_rp = var_weighting_func(lr_grads, rp_grads)
119     else:
120         k_lr, k_rp = message_in['k_lr_k_rp']
121
122     outputs = Listed(log_prob, output)
123     grads = Listed(k_lr * lr_grad_outputs, k_rp * detached_output.grad)
124     message_grads = {'outputs': outputs, 'grads': grads, 'targets': -1}
125
126     message_k = {
127         'k_counter': k_counter + 1,
128         'k_lr_k_rp': (k_lr, k_rp),
129         'targets': 0
130     }
131
132     if k_interval > 1:
133         messages = (message_grads, message_k)
134     else:
135         messages = message_grads
136     return messages
```

Listing 6: Example prototype python code for the backward methods of the propagators

**Baseline functions.**

```
1  # baseline_funcs.py
2  import torch
3
4
5  def mean_baseline(local_losses, node=None):
6      with torch.no_grad():
7          batch_size = local_losses.numel()
8
9          # leave-one-out baselines
10         sum_loss = local_losses.sum()
11         # The division by (batch_size - 1) instead of (batch_size)
```

```
12            # is algebraically equivalent to a leave-one-out baseline.
13            sum_loss = (sum_loss - local_losses) / (batch_size - 1)
14            losses = local_losses - sum_loss
15        return losses
16
17
18  def no_baseline(local_losses, node=None):
19      return local_losses
```

Listing 7: Example prototype python code for the baseline methods used in LR-based MC gradient estimators

## Utilities.

```
1  # utils.py
2  import torch
3
4
5  def expand(data, batch_size):
6      """ Returns the tensor with batch dimension expanded.
7
8      Arguments:
9          data (torch.Tensor): input tensor
10         batch_size (int): batch size for expansion
11
12     Returns:
13         torch.Tensor: output tensor
14
15     """
16     return data.expand((batch_size, ) + data.shape)
17
18
19  def inverse_variance_weighting(x1, x2, scalar_estimate=True):
20      """ Returns weights of inverse variance weighting for each input.
21
22      """
23
24      if isinstance(x1, (list, tuple)) and isinstance(x2, (list, tuple)):
25          x1_vars = []
26          x2_vars = []
27          c_list = []
28          for v1, v2 in zip(x1, x2):
29              assert v1.shape == v2.shape
30              batch_size = v1.shape[0]
31
32              d1 = v1 - v1.mean(dim=0, keepdims=True)
33              d2 = v2 - v2.mean(dim=0, keepdims=True)
34
35              c1 = torch.max(torch.abs(d1))
36              c2 = torch.max(torch.abs(d2))
37              c = torch.max(c1, c2)
38              c_list.append(c)
39
40              if c == 0:
41                  x1_vars.append(torch.tensor(0.0, device=c.device))
42                  x2_vars.append(torch.tensor(0.0, device=c.device))
43              else:
44                  x1_vars.append(torch.sum((d1 / c)**2))
45                  x2_vars.append(torch.sum((d2 / c)**2))
46          x1vec = torch.tensor(x1_vars)
47          x2vec = torch.tensor(x2_vars)
48          cvec = torch.tensor(c_list)
49          cmax = torch.max(cvec)
50          if cmax == 0:
51              x1_var = torch.tensor(1.0, device=cmax.device)
52              x2_var = torch.tensor(1.0, device=cmax.device)
53          else:
54              cvec = (cvec / cmax)**2
55              x1_var = torch.sum(x1vec * cvec)
56              x2_var = torch.sum(x2vec * cvec)
57      else:
58          assert x1.shape == x2.shape
59          batch_size = x1.shape[0]
60
61          d1 = x1 - x1.mean(dim=0, keepdims=True)
62          d2 = x2 - x2.mean(dim=0, keepdims=True)
63
64          c1 = torch.max(torch.abs(d1))
65          c2 = torch.max(torch.abs(d2))
```

```
66        c = torch.max(c1, c2)
67
68        if c == 0:
69            x1_var = torch.tensor(1.0, device=c.device)
70            x2_var = torch.tensor(1.0, device=c.device)
71        else:
72            x1_var = torch.sum((d1 / c)**2)
73            x2_var = torch.sum((d2 / c)**2)
74
75    k_x1 = x2_var / (x1_var + x2_var)
76    k_x1 = torch.clip(k_x1, 0, 1)
77
78    if torch.isnan(k_x1):
79        print('Warning: estimated k was nan. Automatically changed to 0.5.')
80    k_x1[torch.isnan(k_x1)] = 0.5  # when 0/0 error occurs, take them equally.
81
82    return k_x1, 1.0 - k_x1
```

Listing 8: Example prototype python code for utility functions used in Proppo