# OpenReview forum: "Proppo: a Message Passing Framework for Customizable and Composable Learning Algorithms"
_NeurIPS.cc/2022/Conference — NeurIPS 2022 Accept_

### Official Review · Reviewer_GmXe · 2022-07-09

**Rating:** 2
**Confidence:** 1
**Soundness:** 1 poor
**Presentation:** 2 fair
**Contribution:** 1 poor

**Summary:**

Existing automatic differentiation (AD) frameworks do not provide flexibility for composing learning algorithms. This paper invents Automatic Propagation (AP) software, which generalizes AD, and allows the construction of complex learning algorithms.



**Questions:**

See "Strengths And Weaknesses"

**Limitations:**

See "Strengths And Weaknesses"

**Strengths And Weaknesses:**

This work serves as, in my opinion, a kind of software illustration that incorporates modern learning algorithms. As a result, I didn't find this paper to make a major academic contribution.

---

> ### Author Response · Authors · 2022-08-02
> **Response**
>
> Indeed, science vs engineering is a hot debate in our community.
>
> We note that:
>
> 1. There are several precedents of software papers being published at NeurIPS and other top conferences. Such contributions are highly valued in the machine learning community. The NeurIPS call for papers also includes: “Infrastructure (e.g., datasets, competitions, implementations, libraries)”
>
> 2. Our paper is not a pure engineering contribution. The invention of a new class of software is a novel scientific contribution. For example, while modern implementations of AD software lean towards engineering contributions, the first work to introduce AD software was a strong scientific contribution. We view our invention of automatic propagation software as being more similar to the latter, rather than mainly an engineering contribution. Moreover, our work also includes experiments that are of independent interest.

---

### Official Review · Reviewer_JEb5 · 2022-07-09

**Rating:** 3
**Confidence:** 4
**Soundness:** 3 good
**Presentation:** 2 fair
**Contribution:** 2 fair

**Summary:**

The authors present a software framework (Proppo) for constructing propagation algorithms on generic computational graphs. These can be used to implement a variety of learning approaches. The authors make the case that existing ML frameworks, because of their limited capabilities of operating on generic computation graphs, discourage researchers from pursuing particular directions. The authors present example approaches to Monte Carlo gradient estimator and the Total Propagation algorithm implementations using Proppo, then show an experiment demonstrating Total Propagation can improve stability of an RNN-based system with nearly-chaotic dynamics.


**Questions:**

> implemented entirely in Python

Given that the entire framework is implemented in Python, how does performance hold up for large graphs? Large graphs emerge in many learning paradigms (e.g. transduction, graph learning). Performance benchmarks are missing from the paper.

> [Messages are implemented via] a pointer to a parent node

More implementation details regarding statements like this would be very helpful.

> propagation managers working in parallel

How is this accomplished? Threading? Efficient graph traversal?

**Limitations:**

Proppo as a framework allows users to implement RP and LR algorithms as discussed, but these algorithms are not new, and the authors do not detail why an implementation enabled by Proppo is any better than existing implementations of these algorithms, beyond positing that there may be correctness issues with existing implementations that Proppo helps avoid. These ideas are very underspecified.


**Strengths And Weaknesses:**

##### Strengths

The idea of developing a generic framework to perform propagation on computational graphs is important and indeed opens the doors for many new learning algorithms that may not be based on gradient-based learning or may involve gradient computation with higher order or other methods. There is a need for such a generic framework to implement such algorithms.

##### Weaknesses

The paper is very light on the details of the framework itself — the APIs, semantics for implementing other learning algorithms, or implementation details are not explained at all in neither the paper nor the appendix. This makes verifying and validating the quality and correctness of the authors’ implementations difficult.

The work is missing a set of performance benchmarks, which are especially relevant for Monte Carlo-based estimation methods. Total Propagation or similar methods are not new, and while an implementation of them using a generic framework has utility, it is not clear if such an implementation is competitive computationally with other implementations.

The authors present no new learning algorithms imagined with Proppo, instead benchmarking existing ones. The paper would be made stronger if the authors were able to present case studies demonstrating the utility of their approach in more detail. The Monte Carlo gradient estimation example in the paper, while an interesting setting in which to study stability, is not convincingly broadly useful. The authors should make such a case and should expand the set of learning settings in which they explore these algorithms, taking advantage of their interoperability with PyTorch.

---

> ### Author Response · Authors · 2022-08-02
> **(2/2) Response**
>
> Answers to points in the review (continued)
> ____________
>
> **Performance benchmarks:** Perhaps you missed the computational time experiments in App. B.3 (previously B.2), which showed that the computational overhead associated with Proppo is negligible? We realize we had not referenced these experiments in the main paper, so they may have been difficult to spot. We have now added Section 3.5. in the main paper that briefly summarizes the computational aspects, and links to the experiments in the appendix. In the appendix, we compared the computational times of the MC gradient estimators implemented in Proppo. We also included an implementation of reparameterization gradients without Proppo, so this allows directly comparing with a standard implementation that does not rely on Proppo. The main result was that for very small problem sizes, the overhead from Proppo can be significant, but for moderate problem sizes, the overhead is negligible. One of the points we also tried to emphasize in the paper was that Proppo allows implementing algorithms that could not be otherwise easily implemented. Moreover, the computational time may depend more on the implementation of the underlying algorithm than on whether Proppo is used or not (as Proppo mainly just manages the computations performed by some underlying efficient computation tool such as PyTorch). For example, while Proppo allows implementing algorithms that go beyond the surrogate loss formalism, if desired, one can also implement algorithms with the typical surrogate loss method, and merely use Proppo as the user interface. In such a case, the computation time would be roughly the same as the implementation without Proppo using a surrogate loss.
>
> We also note that we have tried Proppo in practical model-based reinforcement learning in extensive experiments in our concurrent work (see the data from the Anonymous paper in the supplementary that was cited in the main paper), and the computation time is reasonable, i.e. total propagation typically requires less than 1.5 times the computation time of using regular RP gradients without Proppo, and provides improvements in performance. These experiments included tasks with Gaussian processes in model-based RL, as well as visual model-based RL from pixel observations using neural networks.
>
> **New learning algorithms imagined with Proppo:** We note that in the Anonymous concurrent work, we implement a new learning algorithm Total Propagation X using Proppo. We do not think it would be reasonable to include this new algorithm in the current paper.  We intend the Proppo paper to be primarily about the Automatic Propagation framework itself. As such, there is no space for adding the substantial experiments shown in the supplementary note. Moreover, we believe that the necessary descriptions—a detailed discussion of the TP family of algorithms and the advantage of TPX—would distract from the main objective of introducing automatic propagation software. We believe such derailing of the story would weaken the paper.
>
>
> Answers to Questions
> ___________
> **Is Python slow?** To test the scalability with the number of nodes, we ran a new simple experiment that we have documented in Appendix B.4. We see that there are no issues with handling millions of nodes. The overhead per node for one pass is roughly $10^{-6}$ seconds. Note that usually Proppo itself does not perform any major computations, it merely manages and modifies the computations of some other efficient computational tools achieving good efficiency.
>
> **How are the pointers implemented?** You can either pass the name of the node, or the address of the node, or for example, the target, -1, means to just send the message backwards. The manager will send the message accordingly.
>
> **Do propagation managers work in parallel?** Note that our comment in the figure caption was about general Automatic Propagation software, not about the specific implementation in Proppo. As this may have been a bit confusing, we moved this comment out of the caption into Section 3.5. where we note parallel propagation managers as future work.
>
> Miscellaneous clarifications
> ________________
> >demonstrating Total Propagation can improve stability of an RNN-based system with nearly-chaotic dynamics.
>
> Being a bit pedantic, the stability of the system is the same for all methods; what changes is the stability and accuracy of the gradient estimation.
>
> **Other:**
> Please also see our general response discussing the significance of the work, and the relevance to MC gradient estimation. We hope this discussion may point out other positive facets of our work.
>
> We are looking forward to a productive discussion in the next phase of the review process.

---

> ### Author Response · Authors · 2022-08-02
> **(1/2) Response**
>
> Thanks for the review.
>
> Regarding concrete additions we have made to address the review’s concerns: we ran a new simple experiment to show that Proppo scales to millions of nodes; we included 9 pages of experimental data in the supplementary from our concurrent work to show that Proppo can be used in practical problems (this work was cited as Anonymous in the original paper, and the data is provided merely for reference; it will not be included in the paper); we provided more implementation details in the form of pseudocode in App. B.2 about the propagators with the aim of better illustrating the composability and customizability. We added the Section 3.5. to more prominently highlight the benchmarking experiments we performed in Appendix B, and to discuss computational issues.
>
> Before we address the specific concerns in the review, we feel there may have been some misunderstanding, so we first want to clarify one point.
>
> The review states:
> > Total Propagation or similar methods are not new, and while an implementation of them using a generic framework has utility, it is not clear if such an implementation is competitive computationally with other implementations.
>
> Note that, in practice, there do not exist any implementations of these algorithms to compare against. There are no known implementations besides the original implementations in MATLAB by Parmas in 2018 (that are not publicly available), and there is no record of anyone having been able to reproduce his implementations (despite significant interest in this work together with several follow up works). There is no known sensible method to implement these algorithms using modern machine learning tools. This is an example of what we illustrated in Figure 1, where writing code for one specific application is much simpler than creating code that is reusable. In our work, we propose a solution to this issue. One of our main motivations was to provide a means to implement such algorithms. Also note that there is no competition in terms of computational speed---our implementation can run on a GPU and is much faster.
>
> Answers to points in the review
> ____________
>
> **Details about the framework --- APIs, semantics, etc.:** Here we want to explain our philosophy on what kind of details we want to include in the paper. As we will be releasing the code upon publication, the specific implementation details will be clear, so we do not see value in explaining the minor implementation details. Neither do we intend our paper to be a user manual for Proppo. The main scientific contribution of our submission is the invention of the class of Automatic Propagation software. Proppo is just one implementation of AP software---there could also be other ways to implement it---and, while valuable, we do not consider our code itself the main scientific contribution. We also envision that the code will continue to evolve after publication. For these reasons, we want to focus our discussion in the paper on the main principles of AP software---the parts that will stay constant---and to give illustrative examples to show how the propagator approach leads to composable and customizable machine learning software. That said, we realize there could be a bit more details about the propagators to better illustrate our points. For this reason, we have added section B.2. in the appendix with pseudocode discussing implementation details of the propagators, and how these are composed together using the sequence propagators.

---

### Official Review · Reviewer_KJqN · 2022-07-12

**Rating:** 7
**Confidence:** 3
**Soundness:** 4 excellent
**Presentation:** 4 excellent
**Contribution:** 3 good

**Summary:**

This paper introduces a class of software called Automatic Propagation (AP), an extension of autodifferentiation software as well as a specific instance of it called Proppo. AP allows users to implement custom learning algorithms into the base unit of a "propagator" that can be reused. The authors note that many ML software packages adhere to the same paradigm of a surrogate loss to be used with backpropagation, while other forms of calculations are not optimized for or considered as they are not the standard. As such, they design AP to be a general message passing framework for ML.  The basic unit of Proppo is a propagator that implements both a forward mode (which performs computation, and stores any necessary data for the backward mode ) and backward mode of computation, which performs backward computation sending and recieving messages to other nodes. The whole process is managed by a propagation manager.

The utility of Proppo and the AP general system is demonstrated by the implementation of total propagation and gaussian shaping gradient algorithms for MC gradient estimation, which are then shown to perform (in terms of gradient accuracy) orders of magnitude better than the standard surrogate loss + autodiff method for certain classes of problems.

**Questions:**

In the minimalistic code example section starting on line 163, it could be useful for some readers to understand how Proppo configuration code could be modified to do the default pytorch behavior (normal backpropagation) (I acknowledge that you introduce the BackPropagator later but it might provide some clarity to new readers if mentioned in the first minimal code section that you could simply also use backprop in addition to changing the behavior of the code completely.

I suggest rewording the sentence on 167-169, as it is confusing what the intention is due to the grammatical errors.

If you could open source some code for Proppo, it would make an even more impactful contribution.

**Limitations:**

The authors have addressed some limitations of the work (i.e. that it is ultimately up to the user to use the framework well to achieve the desired aims of composability and customisability. Giving further guidelines and documentation for users on the best ways to use the framework would be worthwhile, especially if open sourced.

**Strengths And Weaknesses:**

Strengths: The paper is well written and motivated. The contribution is important and original. The experimental results are thoughtful and validate the claim that there are situations where total propagation algorithm can be more accurate gradient estimator than those that can be achieved by standard AD software.

Weaknesses: The paper does not seem to link to any open sourced code, which would be a nice addition to the paper, to see some examples of how the more sophisticated propagators could be implemented.

---

> ### Author Response · Authors · 2022-08-02
> **Response**
>
> Thanks for the review.
>
> **Regarding open sourcing the code:** We confirm that we will be releasing the code on github upon publication. We hope to get the community involved in developing the code. We also plan on continuing to use the code in our own future research, to maintain and continually improve it, as well as the documentation.
>
> **Further guidelines, examples and documentation**
> We have added pseudocode for several propagators into Appendix B.2 to better illustrate how the propagators can be constructed, and we link to these explanations in Section 3.4. Please let us know if these are helpful. We will also release documentation together with the code.
>
> **Suggestions on clarifications:** Thanks for the suggestions, we have made changes in the paper that you can see in the updated submission. Please let us know if these changes were satisfactory. We are happy to add any other small clarifications as requested.
>
> **Regarding limitations:**
> We quote the review:
> >The authors have addressed some limitations of the work (i.e. that it is ultimately up to the user to use the framework well to achieve the desired aims of composability and customisability.
>
> We appreciate that our discussion of these limitations was mentioned. However, to give a more balanced view to the reader of the review, we want to add that Proppo includes features such as composable propagator classes that facilitate and simplify implementing composable algorithms. We believe that the introduced paradigm of propagators is a particularly fruitful approach for constructing software with these desired properties.
>
> **Other:**
> Please also see our general response discussing the significance of the work, and the relevance to MC gradient estimation. We hope this discussion may point out other positive facets of our work.
>
> We are looking forward to a productive discussion in the next phase of the review process.

---

> > ### Comment · Reviewer_KJqN · 2022-08-09
> > **Thanks for your reply**
> >
> > Thanks very much for your reply, I am happy to hear you are open sourcing as well as adding pseudocode to increase the clarity of the paper. I think my score is still an adequate representation of my reflection of the paper as it stands, so will be keeping the score.
> >
> > Best of luck!

---

### Author Response · Authors · 2022-08-02
**(2/2) General response. Significance of total propagation, and guaranteed deliverables**

To put our contribution with the new experimental results with TP into perspective, we explain TP a bit, then introduce two related works. For reference, TP was the first work to suggest linearly combining LR and RP gradients with weighting factors k and (1-k), and they proposed a sophisticated scheme for doing this adaptively at a step-wise level in the computation graph. They also mentioned a naive way to do the combination, not at a step-wise level, but to simply compute the gradients separately and combine them in the end. The naive way is in fact a special case of TP. However, the theoretical bound on the improvement using the naive method would be 2 times the best between LR and RP, so it is not very promising. Following the TP work, the following references [1, 2] used the naive method.

[1] Metz, L., Maheswaranathan, N., Nixon, J., Freeman, D., & Sohl-Dickstein, J. (2019, May). Understanding and correcting pathologies in the training of learned optimizers. ICML (oral)

This work [1] used the naive method in the context of metalearning to improve the performance.

[2] Suh, H. J., Simchowitz, M., Zhang, K., & Tedrake, R. (2022, June). Do differentiable simulators give better policy gradients?. ICML (outstanding paper award)

The main technical contribution in this work [2] was a modification to the naive method to detect discontinuities that would bias the RP gradient. The basic method is that they construct a confidence interval around the LR gradient, and if the RP gradient is too far from this confidence interval, they rely more on the LR method. However, if discontinuities are not detected, the method is equivalent to the naive method.

Our results demonstrate that the original TP is orders of magnitude better than these highly regarded follow-up works (the naive method is bounded to improve by 2 times, whereas TP shows 100 times improvement). We believe that this is an interesting and significant result. The main issue was that TP could simply not be implemented by anyone, and our submission solves this problem.


Guaranteed deliverables
-------------------------
We list what we are guaranteed to deliver upon publication to give a “lower bound” on our contribution:
1. We will release the code for Proppo, the first prototype AP software.
2. We will release the first usable implementations of total propagation and Gaussian shaping gradient algorithms.
3. The descriptions and ideas in the paper about the new class of Automatic Propagation software.
4. Experimental results showing that total propagation can simultaneously obtain orders of magnitude improvement over both LR and RP gradients, while being robust to chaos and the gradient variance being constant with the dimensionality.

---

### Author Response · Authors · 2022-08-02
**(1/2) General response. Overview of response, and significance in MC gradient estimation**

Thanks for the reviews.

We are glad all reviewers found what we are doing important.

Specific responses to the reviews are given in the directly attached comments.

The appendix is now attached at the end of the main paper submission (this is allowed according to the FAQ).

We are looking forward to a productive discussion with the reviewers.

The main tangible changes we have made are: 1) We added pseudocode about the propagators, 2) We added a small Section 3.5 to highlight the computational aspects and prominently point towards the previously included benchmarking experiments in the appendix, 3) we attached 9 pages of experimental data in model-based RL from our concurrent work as reference (this work was cited as “Anonymous” in our original submission, but we decided to add the data as evidence for the reviewers. Note that this data will not be included in the current submission, it is just provided for reference to prove that our code is practical), 4) we ran another simple benchmarking experiment and showed that Proppo scales to millions of propagation nodes.

We can also confirm that we will be releasing our code online upon publication.

Here we want to give a bit more discussion about the significance of our work. At the end of the discussion, we also summarize what we are guaranteed to deliver upon publication.

The reviews already discussed our main contribution: the invention of automatic propagation software. However, the contribution in terms of Monte Carlo gradient estimation was not discussed much. This is understandable, as the paper is primarily about the AP framework, and thus the explanations about MC gradients were terse. Nevertheless, we find our experimental results interesting and significant, so we give some more background here, and explain the significance.

The main MC gradient estimation methods are reparameterization (RP) and likelihood ratio (LR) gradient estimators. RP is backpropagation-like; it is accurate for smooth functions and scales constantly with the dimensionality; however, it is unstable in long computation graphs. LR, on the other hand, is robust in long computation graphs; however, in practice its variance is large, and increases linearly with the dimensionality. The major breakthroughs in ML have been achieved with RP type gradients, e.g., neural networks including recent language models are trained with gradients, because black box methods, such as LR are inefficient. The pinnacle of MC gradient estimation would be to achieve the best properties of both: to have robust estimation in long computation graphs, constant scaling with the dimensionality, and accurate gradient estimation (orders of magnitude better than LR). Our results show for the first time that this is possible. Total propagation (TP) improves by orders of magnitude (i.e., 100 times) in terms of gradient accuracy simultaneously over LR and RP gradients. Moreover, the experiments with Gaussian shaping gradients in Figure 11 in the appendix show that the combination with GS gives constant variance with the dimensionality even for LR gradients, as well as for TP gradients. Thus the combination of TP and GS achieves the desired properties for ideal MC gradient estimators. We believe this is of great independent interest. To be a bit critical of our results, we have not yet demonstrated that this translates to fundamentally better performance on downstream tasks. Moreover, the results with GS rely on the separability of the computational paths. Nevertheless, such a result has never been demonstrated before, not even on toy problems, and we believe it is of great interest.

---

### Meta-Review · Area_Chair_35ry · 2022-08-31

**Recommendation:** Accept
**Confidence:** Less certain

**Metareview:**

As an AC of the paper, I have read this paper and made the decision not only based on the reviewers' reviews but also considering my own judgment as well. Unfortunately, one reviewer could not provide a review with high confidence, and the other two disagreed where Reviewer KJqN was willing to nominate the paper for publication.

# Summary
This paper presents a software framework for a method that the authors called as "Automatic Propagation" (AP.)  AP generalizes AD and allows us to implement custom and composable learning algorithms. However, most of the ML frameworks previously focused on AD rather than implementing general message passing algorithms for learning with Monte-Carlo gradient estimation. This paper proposes a software framework called Proppo implemented in PyTorch to fill this gap. Proppo has "base units" called *propagators* that implements the forward and backwards mode of the computation. Propagators are quite flexible and users of the framework could easily download propagators to implement algorithms like backpropagation and the propagators are managed by "*Propagation Managers*". The paper uses this software framework (Proppo) to implement algorithms like total propagation and gaussian-shaped gradients. Total propagation is, in particular, interesting because it combines LR and RP. It did not have a python implementation before. The authors provide extensive analysis of those both in terms of variance and the speed of their framework.

# Strengths and Weaknesses
This paper studies an important and interesting problem in machine learning on designing software frameworks that could easily enable the implementation of learning algorithms that make use of Monte Carlo gradient estimation. I am not an expert on message-passing frameworks, but the overall proposal looks novel to me. Overall, it feels like this paper's strengths outweigh the weaknesses. I think majority of the weaknesses could be easily addressed in the camera-ready version of the paper.

**Strengths**

* Important problem and provides a well-thought solution for automatic propagation.
* Thorough and convincing experiments.
* Provides a simple and flexible solution for AP, which I hope will enable future research by enabling the implementations of MC gradient estimation algorithms like total propagation that didn't have a python implementation before.

**Weaknesses**

* *Clarity:* The writing could be improved; as Reviewer JEb5 noted, the paper is very light on the details of the framework. Figure 1 feels unnecessary, and the same story could be told without it. Figure 2 is not referred from anywhere in the main text. That figure lacks sufficient description and is difficult to understand. For example, what do the dashed lines in Figure 2 mean?
In the paper, it is not clear what makes an algorithm like "Total Propagation" to be difficult to implement in AD-based frameworks, such as in Storchastic framework. The authors should clarify in this paper why it is easier/better to implement these algorithms with AP and Proppo. Especially it feels like the authors can do a better job at justifying and motivating message passing algorithms in the paper. In the current version, this is briefly done in the intro by citing Minka et al., 2019 slides.
* *Code* The current version of the paper lacks the source code, but the authors promised to release the code with the paper. I would at least expect to see a submission of code with the supplementary material for an early review. Especially it would be very helpful if the authors could present an example of an ML model such as a variational auto-encoder implemented with Proppo.

## Decision
The paper and the software framework Proppo are important contributions to the community that might interest the broader NeurIPS community. As far as I am aware, AP proposed in this paper is novel, and it helps implement Monte-Carlo gradients on stochastic computational graphs. There are also a bunch of other possible use-cases of Prop the authors mentioned. I can definitely see this type of framework becoming popular quickly in the community if it is implemented well and is easy to use. From the examples provided in the paper, Proppo seems to have a relatively simple interface and is easy to use. I am willing to nominate this paper for acceptance with the condition that the framework will be open-sourced with the publication of this paper. Without open-sourcing Proppo this paper would not be as impactful. Besides, I would recommend the authors clarify the following points in the camera-ready version of the paper:
* Refer to the computational time experiments in App. B.3 (previously B.2) from the main paper.
* The paper should incorporate the clarifications requested by reviewer KJqN into the main paper.
* The paper should compare and clarify its differences with respect to other stochastic gradient estimation PyTorch libraries such as  Storchastic.
* I would recommend that the authors address my concerns about the clarity and code in the weaknesses section for the camera-ready version of the paper.



**Award:**

No

---

### Decision · Program_Chairs · 2022-09-14

Accept